# A Multi-Institutional Multimodal EEG Benchmark for Foundation Model Generalization and Early Neurological Diagnosis

## Abstract

Recent advances in deep learning have accelerated the development of foundation models (FMs) for electroencephalography (EEG), with significant efforts devoted to assembling EEG datasets and training large-scale models. However, existing EEG datasets remain highly fragmented and non-standardized, with limited regional diversity since most originate from the United States. Similarly, current EEG foundation models are trained on different datasets without consistent protocols, making it difficult to compare architectures fairly. Moreover, most existing models are trained exclusively on unimodal EEG signals, limiting their clinical utility, as many downstream diagnostic tasks, such as detecting neurodegenerative diseases, require integration of additional modalities beyond EEG. To address these limitations, we introduce, for the first time M-EEG, a multimodal EEG dataset comprising over 6000 patients collected from two major hospitals outside the US. In parallel, we unify several key public EEG datasets into a single standardized corpus, enabling the first rigorous benchmarking of state-of-the-art EEG foundation model architectures under consistent pretraining and fine-tuning pipelines. Finally, we configure and evaluate multimodal diagnostic models based on existing EEG foundation architectures, demonstrating that integrating auxiliary modalities (e.g., blood biomarkers and clinical notes) with EEG substantially improves downstream prediction accuracy, for instance, achieving a 27.64% gain in Alzheimer's disease risk prediction.

## 1 Introduction

**Background.** Recent breakthroughs in deep learning have catalyzed the development of foundation models (FMs) for electroencephalography (EEG) Wang et al. (2025; 2024a;b); Yang et al. (2023); Kostas et al. (2021), with the goal of learning transferable neural representations across diverse clinical and cognitive tasks. In parallel, efforts have been made to assemble large-scale clinical EEG corpora from multiple hospitals (Khan et al., 2022; Zhang et al., 2018; Sun et al., 2025), aiming to broaden regional and clinical diversity and to better capture the inherently non-stationary, low signal-to-noise characteristics of EEG. Despite these encouraging developments, existing EEG datasets and foundation models continue to face significant limitations.

**Limitations of existing EEG datasets and foundation models.** On the dataset side, available corpora remain fragmented: most are heavily US-centric (Obeid & Picone, 2016; Sun et al., 2025), task-specific (Zhang et al., 2018), or involve relatively few subjects (Khan et al., 2022). Such constraints exacerbate overfitting when applying self-supervised pretraining methods, such as mask prediction (Wang et al., 2024b;a; 2025; Yang et al., 2023) or contrastive learning (Yang et al., 2023; Kostas et al., 2021), which depend critically on a wide subject pool to generate reliable positive and negative pairs. Moreover, most datasets lack integration with minimally invasive modalities such as blood-based biomarkers, which could be combined with EEG to strengthen diagnostic accuracy. The recently introduced Harvard Electroencephalography Database (Sun et al., 2025) partially addresses these limitations by releasing nearly three million hours of data from four hospitals, yet it remains entirely US-based and thus insufficient for studying regional diversity at scale.

Concerning the EEG foundation models, current models (e.g., EEGPT(Wang et al., 2024a), BIOT(Yang et al., 2023), CBraMOD(Wang et al., 2025)) exhibit two fundamental limitations: limited regional diversity and restricted clinical relevance. First, most models are pretrained on only a handful of public datasets—largely from single regions, resulting in poor generalization across demographic, clinical, and recording variations. Performance drops sharply when evaluated on unseen regional datasets, underscoring their vulnerability to distribution shifts (See Fig. 3). Dataset heterogeneity in sampling rates, channel layouts, and annotation protocols further complicates the establishment of standardized pretraining pipelines, reinforcing the need for a harmonized and globally representative corpus. Second, existing foundation models are trained exclusively on unimodal EEG signals, whereas real-world diagnosis of complex brain disorders, such as Alzheimer's disease, often requires multimodal integration, including minimally invasive biomarkers like blood-based tests. As illustrated in Table 7, incorporating auxiliary signals substantially improves disease prediction performance over EEG alone, reinforcing the need for multimodal foundation modeling. Yet, there remains a scarcity of public EEG datasets that are both regionally diverse and enriched with complementary clinical modalities.

**Our approach.** To address these gaps, we present M-EEG, a large-scale, clinically annotated EEG dataset collected from two major hospitals outside of US, comprising $1,170$ hours of EEG recordings from $6,081$ patients. To our knowledge, this is the largest non-US clinical EEG corpus to date, offering significant improvements in geographic coverage, subject diversity, and clinical complexity. In addition, a unique subset of M-EEG includes paired EEG, blood biomarkers, and clinical notes, enabling the first non-US multimodal benchmark for EEG–lab fusion.

Building on M-EEG, we conduct a standardized benchmarking study of state-of-the-art EEG foundation models under identical pretraining and fine-tuning protocols across diverse clinical tasks drawn from both US-based and non-US datasets. Our findings demonstrate that pretraining on M-EEG yields stronger generalization across regions and diseases, with clear gains on challenging diagnostic tasks such as early Alzheimer's risk prediction.

**Contributions.** Our contributions are summarized as follows:

- **M-EEG: a large-scale multimodal EEG corpus.** We release M-EEG, a large-scale clinical EEG corpus with 1,170 hours from 6,081 patients at two major hospitals, marking the largest non-US EEG dataset by subject count and improving the diversity of EEG pretraining resources. Furthermore, we curate a subset of M-EEG that integrates EEG signals with blood-based biomarkers and clinical notes, establishing the first non-US multimodal EEG benchmark and opening new avenues for research in EEG-laboratory data fusion. In addition, we standardize multiple existing EEG datasets to construct a unified large-scale corpus and establish a benchmark to compare state-of-the-art EEG foundation model architectures on this dataset. To the best of our knowledge, this is the first standardized large-scale EEG corpus, and our work represents the first systematic benchmarking of EEG foundation models on a common dataset using consistent pretraining and fine-tuning pipelines, thereby enabling rigorous and dataset-independent comparison.

- **Multimodal benchmarking of EEG foundation models for neurological diagnosis.** We adapt existing EEG foundation architectures to a multimodal setting for neurological disorder diagnosis, enabling benchmarking of their performance when combined with additional modalities. Experiments conducted on our curated multimodal EEG dataset, validated through Alzheimer's risk prediction and the diagnosis of epilepsy, transient ischemic attack (TIA), and Parkinson's disease, demonstrate that incorporating additional modalities substantially enhances prediction accuracy.

## 2 EXISTING DATASETS AND EEG FOUNDATION MODELS

### 2.1 CURRENT PRETRAINING CORPORA

Table 1 provides an overview of major EEG datasets used for representation learning, emphasizing their scale, geographic coverage, and any multimodal extensions. The field currently relies on a patchwork of hospital-based clinical EEG corpora as the backbone for foundation model pretraining.

Foremost among these is the Temple University Hospital (TUH) corpus (Obeid & Picone, 2016), which at roughly 24,000 hours of recordings from a single US hospital has underpinned much of the recent progress in self-supervised EEG representation learning (Wang et al., 2025; Han et al., 2025). More recently, the Harvard Electroencephalography Database (HEEDB) (Sun et al., 2025) introduced an unprecedentedly large corpus on the order of millions of EEG hours, drawn from multiple US hospitals and enriched with patient metadata and auxiliary modalities, integrating demographics, medication records, lab values, and free-text clinical notes (including blood-based biomarkers). This rich multimodal resource significantly expanded data scale and scope; however, it remains entirely US-based, exacerbating a persistent regional diversity gap in EEG data. Beyond the United States, only a few smaller clinical corpora have been released. For example, the NMT-Scalp dataset from Pakistan (Khan et al., 2022) provides valuable clinical EEG data but remains limited in scale, with relatively few hours and subjects compared to TUH or HEEDB.

In addition to clinical datasets, a variety of laboratory or task-specific EEG datasets have been used for representation learning. Notable examples include SEED (Zheng & Lu, 2015) for emotion recognition, PhysioNet MI (Goldberger et al., 2000) for motor imagery, M3CV (Huang et al., 2022) for cognitive workload, HGD (Schirrmeister et al., 2017) for brain-computer interface trials, and SHHS (Zhang et al., 2018) for sleep monitoring. While each contributes valuable data for its specific domain, these datasets are relatively small in scale (often involving only tens of subjects or a few dozen hours) and narrow in clinical scope. Moreover, they are typically single-modality (EEG only) and collected under disparate protocols.

## 2.2 EXISTING EEG FOUNDATION MODELS

### 2.2.1 UNIMODAL EEG-BASED FOUNDATION MODELS

EEG foundation models aim to learn general-purpose neural representations from large corpora without relying on task-specific labels. Table 8 summarizes representative architectures and their original pretraining data.

Two open-source efforts, **BENDR** (Kostas et al., 2021) and **CBraMOD** (Wang et al., 2025), were trained exclusively on the TUH clinical corpus, leveraging the breadth of U.S. hospital EEG recordings to drive self-supervised learning objectives. These works established TUH as the standard backbone for EEG foundation modeling. By contrast, **EEGPT** (Wang et al., 2024a) expanded beyond a single corpus by pretraining on a composite of multiple laboratory datasets, including PhysioNet MI, SEED, M3CV, HGD, and TSU to capture a wider spectrum of motor imagery and cognitive tasks. Similarly, **LaBraM** (Jiang et al., 2024) aggregated a heterogeneous collection of public corpora (e.g., TUEG subsets, BCIC IV-1, EmoBrain, Inria BCIC, SPIS Resting) together with private data, aiming to maximize training diversity through scale and variety. Another line of work has drawn on large-scale clinical cohorts beyond TUH. **BIOT** (Yang et al., 2023), for instance, leverages both SHHS, a population-level sleep study, and a small subset of HEEDB collected at Massachusetts General Hospital to pretrain a transformer architecture designed for cross-dataset generalization. Unlike models tied to narrowly defined tasks, BIOT emphasizes scalability across heterogeneous clinical EEG corpora, though its training sources remain limited to US-based datasets (with only a small subset of HEEDB included).

Despite their architectural differences and varying objectives, a common limitation is that each foundation model was developed using a distinct, and often narrow, pool of pretraining data. This inconsistency makes reported improvements difficult to attribute: performance gains may arise as much from the scale, scope, or bias of the underlying corpus as from innovations in model design. Consequently, direct comparison across models remains problematic without a unified and standardized pretraining benchmark.

### 2.2.2 TOWARD MULTIMODAL EEG FOUNDATION MODELS

In clinical practice, EEG is rarely interpreted in isolation. Neurologists routinely contextualize EEG findings with additional information such as blood biomarkers (indicating infection, inflammation, or metabolic abnormalities), routine laboratory test results, and clinical notes that capture patient history and diagnostic impressions. In many neurological disorders, further confirmation may require complex and costly procedures such as MRI, which highlights the value of minimally invasive

Table 1: **Existing EEG pretraining corpora.** BBB denotes blood-based biomarkers. Dataset names are color-coded as follows: blue for general clinical EEG corpora, brown for task-specific corpora, and **bold** for our contribution (**M-EEG**).

| Dataset name | Region | # Hours | # Subjects | # Sites | # Channels | Sampling (Hz) | Modalities | |
|---|---|---|---|---|---|---|---|---|
| | | | | | | | BBB | Clinical notes |
| HEEDB (Sun et al., 2025) | US | 3 000 000 | 109 178 | 4 | 22–57 | 200–512 | ✓ | ✓ |
| TUEG (Obeid & Picone, 2016) | US | 24 000 | 10 874 | 1 | 31 | 250–256 | ✗ | ✗ |
| NMT Scalp (Khan et al., 2022) | Pakistan | 625 | 60 | 1 | 19 | 200 | ✗ | ✗ |
| M3CV (Huang et al., 2022) | China | 90 | 106 | 1 | 64 | 250 | ✗ | ✗ |
| SEED series (Zheng & Lu, 2015) | China | 200 (total) | 8–20 | 1 | 62 | 1000 | ✗ | ✗ |
| PhysioNet MI (Goldberger et al., 2000) | US | 47 | 109 | 1 | 64 | 160 | ✗ | ✗ |
| Inria BCIC (Margaux et al., 2012) | France | 30 | 26 | 1 | 56 | 200 | ✗ | ✗ |
| BCIC IV-1 (Blankertz et al., 2007) | Europe | 8 | 7 | 1 | 59 | 1000 | ✗ | ✗ |
| HGD (Schirrmeister et al., 2017) | China | 15 | 154 | 1 | 128 | 500 | ✗ | ✗ |
| Raw EEG Data (Trujillo, 2020) | US | 34 | 48 | 1 | 64 | 256 | ✗ | ✗ |
| Grasp and Lift (Luciw et al., 2014) | UK | 12 | 12 | 1 | 32 | 500 | ✗ | ✗ |
| EmoBrain (Savran[1] et al., 2006) | Germany | 5 | 16 | 1 | 64 | 1024 | ✗ | ✗ |
| Resting State (Trujillo et al., 2017) | US | 3 | 22 | 1 | 72 | 256 | ✗ | ✗ |
| SPIS Resting (Torkamani-Azar et al., 2020) | China | 1 | 10 | 1 | 64 | 2048 | ✗ | ✗ |
| Target vs Non-Target (Korczowski et al., 2019) | France | 16 | 43 | 1 | 32 | 512 | ✗ | ✗ |
| TSU (Wang et al., 2016) | China | 14 | 35 | 1 | 64 | 250 | ✗ | ✗ |
| SHHS (Zhang et al., 2018) | US | 43 446 | 5 804 | – | 2 | 125 | ✗ | ✗ |
| Siena Scalp (Detti, 2020) | Italy | 30 | 14 | 1 | 29 | 512 | ✗ | ✗ |
| **M-EEG** | Outside of US | 1 170 | 6 081 | 2 | 22–44 | 200, 500 | ✓ | ✓ |

signals that can complement EEG in a more accessible way. These auxiliary data sources provide critical context that can help disambiguate EEG abnormalities and improve diagnostic accuracy.

Despite this reality, most existing EEG foundation models remain strictly unimodal, trained only on raw EEG signals without auxiliary modalities. This limitation reduces their clinical utility: a model that sees only EEG may miss critical disease indicators that would be apparent if combined with supporting evidence such as blood tests or clinical reports.

Extending pretraining corpora beyond EEG is therefore essential for developing foundation models that generalize across diverse clinical scenarios. Incorporating modalities such as blood-based biomarkers and textual clinical records into EEG representation learning can capture patterns more consistent with real-world diagnostic reasoning (Moretti, 2015; Chetty et al., 2024), potentially improving performance on tasks like early detection of neurodegenerative diseases or prognostication after brain injury.

These considerations motivate the collection of multimodal EEG datasets that combine electrophysiological signals with complementary clinical information. In the next section, we present **M-EEG**, a multi-institutional dataset that pairs EEG recordings with blood biomarkers and clinical notes, and introduce a unified benchmarking framework for evaluation. Together, these contributions expand regional coverage, integrate multimodal context, and enable fair, standardized assessment of EEG foundation models.

## 3 MULTI-INSTITUTIONAL MULTIMODAL EEG DATASET

In the following, we introduce a multi-institutional EEG dataset that has been systematically compiled and meticulously curated to support advanced research in computational neuroscience. The dataset comprises three main components.

The primary component is **M-EEG** (Section 3.1), our in-house multimodal dataset collected outside the United States, which includes synchronized EEG recordings alongside corresponding blood test results. This multimodal dataset not only enhances the diversity of existing EEG data populations, thereby improving the generalizability of EEG foundation models (as demonstrated in Section 4.3), but also leverages its multimodal nature to boost performance on downstream tasks, as will be further discussed in Section 4.4.

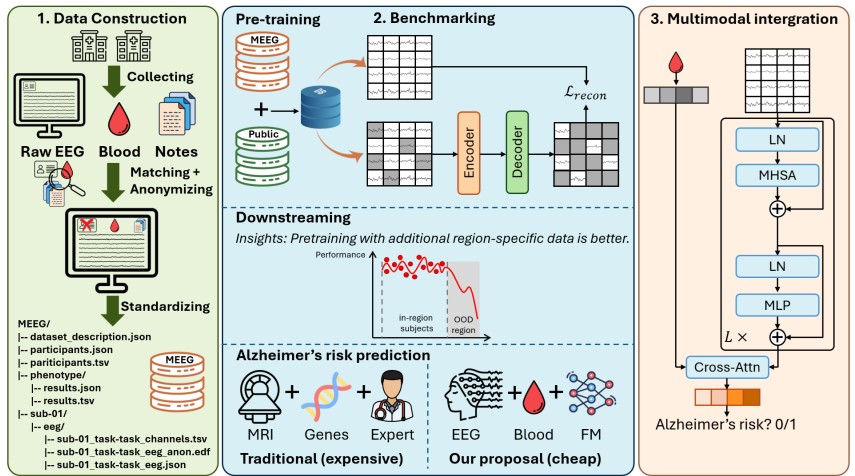

Figure 1: Overview of M-EEG. (1) **Data construction:** raw EEG, blood biomarkers, and clinical notes collected from two hospitals are anonymized and standardized into BIDS format. (2) **Benchmarking:** M-EEG enables large-scale pretraining and standardized evaluation of EEG foundation models, with downstream results showing that region-specific data improves *regional robustness*. (3) **Multimodal integration:** M-EEG includes paired EEG–blood data, allowing exploration of multimodal foundation models for clinical tasks such as early disease risk prediction.

In order to benchmark existing foundation architectures, we further introduce **P-EEG** and **T-EEG**. The **P-EEG** component (Section 3.2) is a unified public dataset constructed through the aggregation and harmonization of multiple publicly available EEG datasets. It is designed specifically for the pretraining of EEG foundation models. By standardizing data formats and preprocessing pipelines, this unified corpus offers a robust, scalable, and reproducible benchmark for training, evaluating, and comparing foundation models in EEG-based machine learning research. Finally, the **T-EEG** component is derived from publicly available task-oriented datasets and is specifically curated to evaluate the performance of foundation models on a range of targeted downstream tasks.

## 3.1 M-EEG: AN IN-HOUSE MULTI-INSTITUTIONAL, MULTIMODAL EEG DATASET

We construct M-EEG, a multi-institutional, multimodal EEG dataset, collected from two major hospitals, namely Hospital A and Hospital B, located outside the United States. The primary objective of this dataset is to enhance the diversity of existing EEG datasets, both in terms of geographical representation (regional diversity) and data modality. As illustrated in Table 2 and Figure 2, the multimodal subset exhibits a diverse age and gender distribution. Moreover, all patients in our dataset are recruited from a country geographically distant from the United States, providing regional characteristics that are complementary to existing US-centric EEG corpora. Using this dataset, we demonstrate that regional diversity plays a critical role in improving EEG representation learning for foundation models, while incorporating additional modalities beyond EEG, such as blood biomarkers, significantly boosts the accuracy of brain-related disease prediction.

The construction of M-EEG involved several key steps: (1) raw data acquisition, (2) cross-modality synchronization, and (3) standardized data preprocessing.

**Raw data acquisition. M-EEG** advances beyond prior corpora by providing the largest non-US clinical EEG cohort to date, comprising $1,170$ hours of routine clinical EEG collected from $6,081$ patients across two hospitals over multiple years. The detail configurations are presented in Table 3. All recordings were fully de-identified before release, with patient identifiers removed and institution-specific metadata anonymized, thereby preserving clinical fidelity while ensuring compliance with privacy and ethical standards.

**Cross-modality synchronization.** In addition to raw EEG, the subset from Hospital B includes paired blood-based biomarkers (BBB) and clinical notes, enabling multimodal representation learning. We report the statistics for this subset in Table 2 and Figure 2. Currently, the dataset contains only single-day EEG recordings per patient, without multi-day follow-up sessions. Laboratory re-

Table 2: Summary of age and gender distribution in the multimodal corpus from Hospital B.

| Year | Patients (M, F) | Age (years) |
|------|-----------------|-------------|
| 2019 | 8 (2, 6) | $62.5 \pm 9.77$ |
| 2020 | 11 (1, 10) | $55.6 \pm 16.12$ |
| 2021 | 20 (3, 17) | $53.5 \pm 17.98$ |
| 2022 | 35 (3, 32) | $73.3 \pm 7.94$ |
| 2024 | 2235 (497, 1738) | $44.09 \pm 17.94$ |
| 2025 | 2795 (850, 1945) | $46.19 \pm 17.87$ |
| Total | 5104 (1356, 3748) | $45.88 \pm 18.08$ |

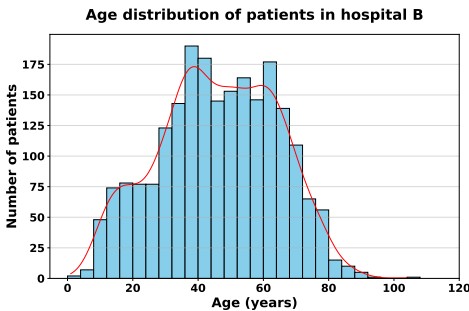

Figure 2: Age distribution of patients in the multimodal corpus from Hospital B

Table 3: Site-specific configurations of 2 hospitals in **M-EEG**.

| | Hospital A | Hospital B |
|---|---|---|
| **# Patients** | 947 | 5,134 |
| **# Records** | 947 | 5,272 |
| **# Hours** | 290 | 880 |
| **Channels** | 22 | 44 |
| **Sampling (Hz)** | 200 | 500 |

Table 4: Performance of EEG foundation models pre-trained on the unified corpus P-EEG and finetuned on task-oriented dataset T-EEG.

| Task | Architecture | Balanced Acc. ↑ | Kappa / AUPR ↑ | W. F1 / AUROC ↑ |
|------|--------------|-----------------|----------------|-----------------|
| BCIC-2a | CBraMOD | 0.4978 | 0.3304 | 0.4856 |
| | EEGPT | **0.5374** | **0.3823** | **0.5138** |
| TUEV | CBraMOD | 0.4449 | 0.5114 | 0.7394 |
| | EEGPT | **0.5217** | **0.5581** | **0.7680** |
| TUAB | CBraMOD | 0.6175 | 0.4384 | 0.6897 |
| | EEGPT | **0.8018** | **0.8800** | **0.8826** |
| Sleep-EDFx | CBraMOD | **0.7512** | **0.7258** | **0.7978** |
| | EEGPT | 0.6585 | 0.5963 | 0.6976 |

sults and clinical notes are collected on the same calendar day as the EEG. The cohort covers a wide spectrum of neurological conditions such as epilepsy, encephalopathy, sleep disorders, and neurode-generative diseases, reflecting real-world clinical diversity. All routine blood-based biomarkers and de-identified clinical notes are centralized in a dedicated `phenotype/` directory. Each patient is linked to two files: `results.tsv`, containing tabular laboratory values, and `results.json`, containing free-text diagnostic notes and impressions.

**Standardization. M-EEG** is organized following the **Brain Imaging Data Structure (BIDS) specification** Gorgolewski et al. (2016), version 1.8.0. At the top level, the dataset is structured according to the BIDS hierarchy, which includes:

- `dataset_description.json`: Contains metadata describing the dataset, its authorship, and BIDS compliance.

- `participants.tsv` and `participants.json`: Contain participant-level demographic and group information.

- `phenotype/`: Contains clinical laboratory test results in `results.tsv` and related metadata in `results.json`.

- `sub-xxxx/`: Contain subject-specific data, including an `eeg/` subfolder with EEG recordings, associated metadata, channel information, and a `sub-xxxx_scans.tsv` file documenting recording timestamps.

### 3.2 P-EEG: A Unified EEG Corpus for Foundation Model Pretraining

To establish a fair and comprehensive benchmark for foundation model pretraining, we aggregate multiple publicly available EEG datasets and integrate them with our proprietary M-EEG dataset to construct a unified corpus, referred to as **P-EEG**, specifically tailored for the training and evaluation of EEG foundation models.

Although a wide range of public EEG datasets exist, each is originally created for distinct research purposes. Therefore, we carefully select only those datasets that align with the objectives and requirements of foundation model training. In the following sections, we detail the criteria used for

dataset selection and describe the preprocessing pipeline employed to harmonize and standardize the selected datasets into a coherent and consistent format.

### 3.2.1 DATASET SELECTION

We selected datasets from Table 1 based on two main criteria: (i) a focus on patient-based clinical recordings rather than task-specific paradigms, and (ii) the ability to ensure both biological and regional diversity while maintaining sufficient EEG channel coverage.

Specifically, we excluded task-oriented datasets, highlighted in brown in Table 1, as they are tailored to narrow cognitive or motor tasks, which can bias representation learning toward predefined downstream objectives. Although the SHHS dataset (Zhang et al., 2018) offers a large sample size, it records only two EEG channels in a sleep-specific context, limiting its applicability for general-purpose pretraining. We also deferred the inclusion of the HEEDB dataset (Sun et al., 2025) due to its massive scale and the ongoing integration process, reserving it for future work.

As a result, the unified dataset, P-EEG, comprises three complementary corpora: the Temple University EEG (TUEG) dataset (Obeid & Picone, 2016), the NMT Scalp EEG dataset from Pakistan (Khan et al., 2022), and our newly introduced dataset, M-EEG. Together, these datasets span multiple hospitals, geographic regions, and acquisition protocols, forming a diverse yet clinically grounded corpus for the training and evaluation of EEG foundation models.

### 3.2.2 DATA PREPROCESSING AND HARMONIZATION

Our preprocessing largely follows CBraMOD (Wang et al., 2025) to reduce variability and remove noise. We discard the first and last minute of TUEG recordings, retain 19 common 10-20 channels, and apply a 0.3-75 Hz band-pass filter plus a 60 Hz notch filter. Signals are resampled at 200 Hz, segmented into 30 s windows, and normalized to $[-1, 1]$ after excluding samples with amplitudes above $100, \mu V$ (Yin et al., 2025). For NMT-Scalp (Khan et al., 2022) and M-EEG, we apply the same pipeline but use a 50 Hz notch filter and Independent Component Analysis (ICA) (Makeig et al., 1995) to further suppress artifacts.

### 3.3 T-EEG: A TASK-ORIENTED EEG BENCHMARK FOR DOWNSTREAM EVALUATION

**Downstream BCI Tasks and Datasets.** T-EEG serves as a task-oriented benchmark designed to systematically evaluate the generalization of EEG foundation models across diverse downstream applications. We include six representative tasks spanning seven EEG datasets, as summarized in Table 9. The benchmark covers well-established challenges in brain-computer interface and clinical EEG analysis: motor imagery (BCIC-2a (Blankertz et al., 2007)), sleep staging (SleepEDF (Kemp et al., 2000)), seizure detection (TUEV (Obeid & Picone, 2016)), and abnormal EEG classification (TUAB (Obeid & Picone, 2016)). To evaluate robustness under regional shifts, we further incorporate A&MISP (Ma Thi et al., 2025), ALS (Ngo et al., 2024), and N-FM (Neurought, 2023), which introduce distinct recording conditions and subject populations. Finally, to assess multimodal integration, we include the external PEARL dataset (Dzianok & Kublik, 2024) for Alzheimer's risk prediction, where paired EEG and blood biomarkers enable evaluation of multimodal representation learning. In addition, we curate three neurological disorder prediction tasks (epilepsy, transient ischemic attack (TIA), and Parkinson's disease) as multimodal subsets of **M-EEG**, where EEG is paired with blood-based biomarkers and/or free-text clinical notes.

**Preprocessing pipeline.** Given the heterogeneity of real-world EEG collections, the datasets in T-EEG vary substantially in sampling frequency, number of channels, and segment duration. To ensure fair comparison, we establish a standardized preprocessing pipeline: linear channel mappings are applied when necessary to align with the pretrained 19-channel montage, and signals are adaptively truncated or segmented around task-specific annotations to extract meaningful samples. Table 9 details the preprocessing setup for each dataset, with further descriptions provided in Appendix A.

## 4 EEG FOUNDATION MODEL BENCHMARKING

In this section, using our dataset, we conduct a series of experiments to address three key research questions: (1) How do state-of-the-art EEG foundation models compare in performance? (Section

4.2); (2) How effective is the M-EEG dataset for pretraining EEG foundation models? (Section 4.3); (3) To what extent does incorporating multimodality improve performance on EEG-related downstream tasks? (Section 4.4).

## 4.1 EXPERIMENT SETTINGS

**Baselines.** We include two state-of-the-art EEG foundation models as baselines. (1) **CBraMOD** (Wang et al., 2025), a reconstruction-based model was originally pretrained on TUH (TUEG). (2) **EEGPT** (Wang et al., 2024a), a multi-corpus model was originally pretrained on laboratory datasets including PhysioNet MI (Goldberger et al., 2000), SEED (Zheng & Lu, 2015), M3CV (Huang et al., 2022), HGD (Schirrmeister et al., 2017), and TSU (Wang et al., 2016).

**Tasks.** We evaluate foundation models on the downstream tasks defined in T-EEG (section 3.3), spanning both multiclass and binary classification settings. More details for each task are described in Appendix A.

**Metrics.** To ensure consistent and interpretable evaluation across tasks, we report performance using metrics tailored to the nature of each dataset. For **multiclass classification** tasks (BCIC-2a, SleepEDF, TUEV, A&MISP, ALS, N-FM), we compute Balanced Accuracy, Cohen's Kappa, and Weighted F1, which account for class imbalance and provide a comprehensive view of classification quality. For **binary classification** tasks (TUAB and PEARL), we report Balanced Accuracy together with AUROC and AUPR, as these metrics are more informative under skewed class distributions.

## 4.2 MODEL COMPARISON

We begin by comparing representative EEG foundation model architectures under a unified pretraining setup. Specifically, all models are pretrained on the P-EEG dataset and then finetuned on the T-EEG dataset.

We report results on four widely recognized tasks, BCIC-2a, TUEV, TUAB, and SleepEDF, spanning distinct BCI tasks, including motor imagery, seizure detection, abnormal EEG classification, and sleep staging. Together, these benchmarks cover both cognitive and clinical applications and provide complementary perspectives on model generalization. Results are summarized in Table 4.

Overall, EEGPT tends to outperform CBraMOD across diverse tasks, likely because its auxiliary alignment loss mitigates mode collapse and yields more discriminative representations, whereas CBraMOD relies solely on masked prediction

## 4.3 IMPACTS OF REGIONAL DATA

As illustrated in Fig. 3, on BCIC-2a, which shares characteristics with the pretraining data described in Table 8, both CBraMOD and EEGPT achieve justifiable performance (balanced accuracy: 0.49 vs. 0.51, Kohen's kappa: 0.32 vs. 0.34, weighted F1: 0.47 vs. 0.49). In contrast, on A&MISP, collected under different regional conditions, performance collapses, with balanced accuracy and F1 reduced by nearly 50% and kappa by more than 95%. To examine regional robustness, we split P-EEG into two subsets: an out-of-region set collected from the same geographic area as M-EEG, and an in-region set collected elsewhere. We then design two experiments: (1) adding M-EEG should not downgrade the performance of models trained on the in-region subset (Table 5), and (2) adding M-EEG should improve the performance of models trained on the out-of-region subset (Table 6).

Table 5 shows that incorporating M-EEG does not degrade performance on the *in-region* subset. Across BCIC-2a, TUAB, and TUEV, most metrics either improve or remain stable. For instance, CBraMOD gains +17.20% balanced accuracy on TUEV and +4.41% on TUAB, while EEGPT improves by +6.39% on BCIC-2a. The few decreases (e.g., EEGPT on Sleep-EDFx, below 3% on secondary metrics) are marginal and do not alter the overall trend. These results confirm that adding M-EEG preserves accuracy on benchmarks that have traditionally anchored EEG foundation model comparisons, ensuring continuity with prior work and demonstrating that regional diversity does not harm in-region tasks.

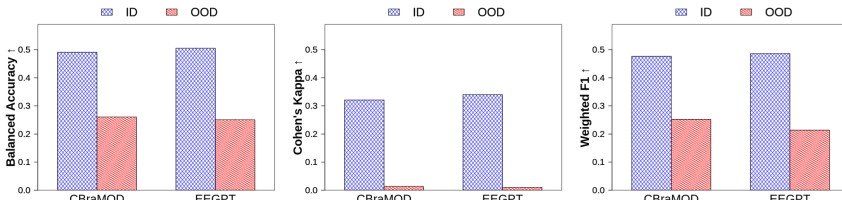

Figure 3: Performance comparison on 4-class motor imagery tasks under in-region (ID) and out-of-region (OOD) settings. BCIC-2a serves as the ID dataset, whereas A&MISP represents the OOD dataset from the region represented in M-EEG.

Table 6 highlights the *out-of-region* subset, where the benefits of M-EEG pretraining are pronounced. Both CBraMOD and EEGPT consistently improve, with substantial relative gains on A&MISP (+8.37% balanced accuracy and +190% Cohen's $\kappa$ for EEGPT) and ALS (+3.74% BA and +19.43% $\kappa$ for EEGPT). Even on the high-performing N-FM dataset, where baselines approach ceiling, CBraMOD achieves a +3.92% improvement in balanced accuracy. These findings show that regional coverage not only maintains comparability on in-region tasks but also directly enhances robustness when models are transferred to populations and recording conditions absent from US-centric corpora.

## 4.4 IMPACTS OF MULTIMODALITY DATA

**Multimodal fusion.** We integrate blood test results with EEG via a simple cross-attention module: blood biomarkers are projected into the EEG embedding space and used as queries to attend over EEG tokens. More details are presented in Appendix A.2. To minimize confounding from lab availability and test-ordering patterns, we focus on subjects sharing a common set of blood tests (see Appendix A.3 and A.4).

### 4.4.1 EXPERIMENTS RESULTS ON PEARL FOR ALZHEIMER'S RISK PREDICTION

**Experiments results.** Table 7 reports Alzheimer's risk prediction on the PEARL dataset across three tasks: MSIT, SMT, and RST. Incorporating blood-based biomarkers alongside EEG consistently improves performance for both CBraMOD and EEGPT. On MSIT, adding BBB yields relative gains of +27.6% balanced accuracy and +37.4% AUPR for CBraMOD, and comparable improvements for EEGPT (+25.1% and +37.6%). Importantly, this +27.6% gain is observed in a setting where the unimodal EEG baseline already achieved balanced accuracy above 0.5, i.e., better than random guessing, underscoring the substantial added value of multimodal integration.

Our preliminary findings demonstrate clear improvements in risk prediction, motivating future work on developing foundation models that seamlessly integrate EEG with other minimally invasive modalities.

### 4.4.2 EXPERIMENTS RESULTS ON M-EEG FOR NEUROLOGICAL DISORDERS PREDICTION

**Experiments results.** Table 7 further reports multimodal risk prediction for epilepsy, TIA, and Parkinson's disease on the M-EEG dataset. Across all three disorders, augmenting EEG with blood-based biomarkers consistently improves performance for both CBraMOD and EEGPT. For epilepsy, multimodal integration yields relative gains of +19.67-22.59% in balanced accuracy, +28.04-40.60% in AUPR, and +26.36-36.35% in AUROC, indicating that BBB features help the models better recover minority-class seizure cases beyond what is achievable from EEG alone.

For TIA, the effect of BBB is particularly pronounced for CBraMOD in terms of AUPR, with a relative improvement of +59.51%, alongside gains of +7.86% in balanced accuracy and +18.74% in AUROC. EEGPT also benefits, though with more moderate improvements, especially in balanced accuracy (+15.00%), AUPR (+7.31%), and AUROC (+7.67%).

Parkinson's disease exhibits the strongest overall performance: with BBB, both architectures reach AUROC values around 0.95 and AUPR above 0.94, together with relative gains of +20.00-24.53% in balanced accuracy, +22.04-23.32% in AUPR, and +16.58-28.70% in AUROC. In summary, the M-EEG experiments corroborate the findings, showing that blood-based biomarkers provide robust,

Table 5: Comparison of EEG foundation models pretrained on the original datasets versus those trained on **P-EEG**, considering datasets from the different regions with M-EEG.

| Tasks | Architectures | | Balanced Acc. ↑ | | Kappa / AUPR ↑ | | W. F1 / AUROC ↑ | |
|---|---|---|---|---|---|---|---|---|
| | | | Perf. | Gain | Perf. | Gain | Perf. | Gain |
| BCIC-2a | CBraMOD | Base | 0.4907 | | 0.3210 | | 0.4766 | |
| | | P-EEG | 0.4978 | +1.45% | 0.3304 | +2.93% | 0.4856 | +1.89% |
| | EEGPT | Base | 0.5051 | | 0.3402 | | 0.4860 | |
| | | P-EEG | 0.5374 | +6.39% | 0.3823 | +12.38% | 0.5138 | +5.10% |
| TUEV | CBraMOD | Base | 0.3796 | | 0.4734 | | 0.7162 | |
| | | P-EEG | 0.4449 | +17.20% | 0.5114 | +8.03% | 0.7394 | +3.24% |
| | EEGPT | Base | 0.5431 | | 0.5361 | | 0.7481 | |
| | | P-EEG | 0.5217 | -3.93% | 0.5581 | +4.10% | 0.7680 | +2.66% |
| TUAB | CBraMOD | Base | 0.5914 | | 0.5685 | | 0.6230 | |
| | | P-EEG | 0.6175 | +4.41% | 0.6167 | +8.48% | 0.6527 | +4.77% |
| | EEGPT | Base | 0.7891 | | 0.8749 | | 0.8708 | |
| | | P-EEG | 0.8018 | +1.61% | 0.8800 | +0.58% | 0.8826 | +1.36% |
| Sleep-EDFx | CBraMOD | Base | 0.7390 | | 0.7316 | | 0.8000 | |
| | | P-EEG | 0.7512 | +1.65% | 0.7258 | -0.79% | 0.7978 | -0.28% |
| | EEGPT | Base | 0.6356 | | 0.6117 | | 0.7062 | |
| | | P-EEG | 0.6585 | +3.60% | 0.5963 | -2.52% | 0.6976 | -1.22% |

Table 6: Comparison of EEG foundation models pretrained on the original datasets versus those trained on **P-EEG**, considering datasets from the same region as M-EEG.

| Tasks | Architectures | | Balanced Acc. | | Kappa | | W. F1 | |
|---|---|---|---|---|---|---|---|---|
| | | | Perf. | Gain | Perf. | Gain | Perf. | Gain |
| A&MISP | CBraMOD | Base | 0.2604 | | 0.0136 | | 0.2523 | |
| | | P-EEG | 0.2715 | +4.26% | 0.0286 | +110.29% | 0.2494 | -1.14% |
| | EEGPT | Base | 0.2507 | | 0.0100 | | 0.2138 | |
| | | P-EEG | 0.2716 | +8.37% | 0.0290 | +190.00% | 0.2234 | +4.49% |
| ALS | CBraMOD | Base | 0.3706 | | 0.1930 | | 0.4047 | |
| | | P-EEG | 0.3715 | +0.24% | 0.2018 | +4.56% | 0.4019 | -0.69% |
| | EEGPT | Base | 0.3448 | | 0.1549 | | 0.3733 | |
| | | P-EEG | 0.3577 | +3.74% | 0.1850 | +19.43% | 0.3843 | +2.95% |
| N-FM | CBraMOD | Base | 0.9192 | | 0.9183 | | 0.9187 | |
| | | P-EEG | 0.9553 | +3.92% | 0.9548 | +3.97% | 0.9551 | +3.96% |
| | EEGPT | Base | 0.9979 | | 0.9979 | | 0.9978 | |
| | | P-EEG | 0.9989 | +0.10% | 0.9990 | +0.11% | 0.9989 | +0.11% |

Table 7: Neurological disorder prediction across PEARL and M-EEG. Alzheimer's risk is evaluated on the PEARL dataset, while epilepsy, TIA, and Parkinson's disease are evaluated on the M-EEG dataset. We compare unimodal EEG (w/o BBB) with multimodal EEG plus blood-based biomarkers (w/ BBB), with teal denoting the relative improvements over the EEG-only baseline.

| Tasks | Architectures | | Balanced Accuracy | | AUPR | | AUROC | |
|---|---|---|---|---|---|---|---|---|
| | | | Performance | Gain | Performance | Gain | Performance | Gain |
| PEARL–MSIT | CBraMOD | w/o BBB | 0.5283 | | 0.5523 | | 0.5877 | |
| | | w/ BBB | 0.6743 | +27.64% | 0.7588 | +37.39% | 0.7779 | +32.36% |
| | EEGPT | w/o BBB | 0.4615 | | 0.4285 | | 0.4063 | |
| | | w/ BBB | 0.5774 | +25.11% | 0.5895 | +37.57% | 0.5976 | +47.08% |
| PEARL–SMT | CBraMOD | w/o BBB | 0.5296 | | 0.4692 | | 0.5040 | |
| | | w/ BBB | 0.6288 | +18.73% | 0.6774 | +44.37% | 0.7156 | +41.98% |
| | EEGPT | w/o BBB | 0.4746 | | 0.4132 | | 0.4222 | |
| | | w/ BBB | 0.5627 | +18.56% | 0.6109 | +47.85% | 0.5651 | +33.85% |
| PEARL–RST | CBraMOD | w/o BBB | 0.4504 | | 0.4445 | | 0.4580 | |
| | | w/ BBB | 0.6960 | +54.52% | 0.7772 | +74.84% | 0.7783 | +69.93% |
| | EEGPT | w/o BBB | 0.4366 | | 0.3925 | | 0.3949 | |
| | | w/ BBB | 0.5753 | +31.77% | 0.5985 | +52.48% | 0.5483 | +38.85% |
| M-EEG-Epilepsy | CBraMOD | w/o BBB | 0.5248 | | 0.4262 | | 0.5142 | |
| | | w/ BBB | 0.6280 | +19.67% | 0.5457 | +28.04% | 0.7011 | +36.35% |
| | EEGPT | w/o BBB | 0.5144 | | 0.4126 | | 0.5494 | |
| | | w/ BBB | 0.6306 | +22.59% | 0.5801 | +40.60% | 0.6942 | +26.36% |
| M-EEG-TIA | CBraMOD | w/o BBB | 0.5266 | | 0.4003 | | 0.6234 | |
| | | w/ BBB | 0.5680 | +7.86% | 0.6385 | +59.51% | 0.7402 | +18.74% |
| | EEGPT | w/o BBB | 0.5446 | | 0.5269 | | 0.5776 | |
| | | w/ BBB | 0.6263 | +15.00% | 0.5654 | +7.31% | 0.6219 | +7.67% |
| M-EEG-Parkinson | CBraMOD | w/o BBB | 0.5556 | | 0.7850 | | 0.7396 | |
| | | w/ BBB | 0.6667 | +20.00% | 0.9681 | +23.32% | 0.9519 | +28.70% |
| | EEGPT | w/o BBB | 0.6157 | | 0.7755 | | 0.8153 | |
| | | w/ BBB | 0.7667 | +24.53% | 0.9464 | +22.04% | 0.9505 | +16.58% |

architecture-agnostic gains across diverse neurological disorders, particularly on clinically challenging tasks.

## 5 CONCLUSION

In this study, we present M-EEG, a novel multimodal EEG dataset collected from two hospitals outside the United States. To support large-scale modeling, we further curated and standardized existing public EEG datasets into two complementary resources: P-EEG, designed for pretraining EEG foundation models, and T-EEG, a suite of task-oriented datasets tailored for finetuning models on specific applications. Leveraging these datasets, we conducted a comprehensive evaluation of the two most advanced EEG foundation models to date. Beyond benchmarking, we also investigate the benefits of pretraining on M-EEG and demonstrate that incorporating multimodal EEG substantially boosts downstream predictive performance across multiple neurological disorders, including Alzheimer's disease, epilepsy, transient ischemic attack (TIA), and Parkinson's disease. In the future, we plan to further enrich **M-EEG** through larger-scale, longitudinal data collection and to explore foundation models that integrate EEG with multiple minimally invasive modalities, aiming toward clinically reliable multimodal foundation models.

## REPRODUCIBILITY STATEMENT

All data used in this study were collected in full compliance with the hospital's internal regulations and ethical guidelines for handling patient and participant information. The dataset employed in this work was provided by the collaborating hospital with explicit authorization for scientific research purposes. With respect to data sharing, the ownership and governance of the original clinical dataset rest with the hospital. Consequently, requests for access to this dataset for research purposes can be directed to the hospital, which will evaluate and share the data in accordance with its regulations, approval procedures, and confidentiality safeguards. If access is approved, the data will be retrieved from a secure cloud environment managed by the hospital (or its authorized provider) and made available only under controlled conditions, ensuring full compliance with data protection, privacy, and security standards. Upon acceptance, we will additionally upload a controlled-access data request form for M-EEG, which researchers can use to request access in accordance with our data-use requirements and the hospital's regulations.

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

# A APPENDIX

Table 8: Summary of recent state-of-the-art architectures for EEG Foundation Models and their original corresponding pretraining data.

| Architectures | Pretraning Datasets |
|---|---|
| CBraMOD | TUEG |
| EEGPT | PhysioNet MI, HGD, TSU, SEED, M3CV |
| LaBraM | a subset of TUEG, BCIC IV-1, EmoBrain, Grasp and Lift, Inria BCIC, Resting State, SPIS Resting, SEED, Siena Scalp, Target vs Non-Target, Raw EEG Data, Private Data |
| BIOT | SHHS, a tiny subset from HEEDB |
| BENDR | TUEG |

## A.1 FINE-TUNING ON DOWNSTREAM TASKS

We load the pre-trained weights of **M-EEG** and replace the reconstruction head with a task-specific head which is composed of multi-layer perceptrons. Here the learned EEG representations are flattened and fed into the task-specific head for downstream tasks. Then we fine-tune **M-EEG** in downstream datasets. **We employ binary cross-entropy (BCE) loss for binary classification, cross-entropy loss for multi-class classification.** More hyperparameters for **M-EEG** fine-tuning on downstream datasets are shown in Table 10. For fair evaluation, we have extensively built a **subject-wise cross-evaluation** scheme, in which all subjects are partitioned into $N$ folds for the validation set or the test set. For example, we conduct N fine-tunings; in each of them, one fold is held out as the test set while the remaining folds are used for training and validation.

### A.1.1 BCIC-2A

**Description & Preprocessing.** BCIC-2A consists of data from 9 subjects doing trials of 4 different motor imagery tasks. These tasks are motor imagery of the left hand (Class 1), right hand (Class 2), feet (Class 3), and tongue (Class 4). Each subject performs two sessions on different days, with each session consisting of 288 trials. We apply a band-pass filter from 0 to 38 Hz, sampling rate at 200 Hz, and 4-second window sample (800 data points).

**Evaluation.** We adopt a leave-one-subject-out (LOSO) cross-validation protocol. We perform 9 fine-tunings, each involving a different subject as a testing dataset, and the remaining 8 subjects serve as the training set. We report the test result of the last checkpoint.

### A.1.2 TUEV

**Description & Preprocessing.** TUEV is a seizure detection dataset, which is a subset of TUEG. This dataset records clinical EEG segments of 6 classes: spike and sharp wave (SPSW), generalized periodic epileptiform discharges (GPED), periodic lateralized epileptiform discharges (PLED), eye movement (EYEM), artifact (ARTF), and background (BCKG). We apply a band-pass filter from 0.1 Hz to 75 Hz and a notch filter at 60Hz, sampling rate of 200 Hz, and 5-second window sample (1000 data points).

**Evaluation.** As TUEV has its own evaluation set, which we regard as the test set. We adopt the proposed cross-validation protocol for validation sets by splitting all subjects into 4 folds. We then conduct 4 fine-tunings, each involving one fold of subjects as a validation set, and the remaining subjects serve as the training set.

### A.1.3 SLEEP-EDFX

**Description & Preprocessing.** Sleep-EDFx is a sleep stage classification dataset, consisting of data from 78 healthy subjects. This dataset contains 5 classes, corresponding to 5 stages of sleep:

Table 9: Summary of T-EEG and its BCI Tasks.

| BCI Task | Dataset | Rate | # Ch. (used) | Duration | # Labels |
|---|---|---|---|---|---|
| Motor Imagery | BCIC-2a | 250 Hz | 22 | 4s | 4 |
| | A&MISP | 128 Hz | 22 | 4s | 4 |
| | ALS | 128 Hz | 19 | 4s | 4 |
| Sleep Staging | SleepEDF | 100 Hz | 2 | 30s | 5 |
| Seizure / Event Detection | TUEV | 250 Hz | 16 | 10s | 4 |
| Abnormal EEG Detection | TUAB | 250 Hz | 16 | 10s | 2 |
| Characters Detection | N-FM | 512 Hz | 1 | 1s | 94 |
| Alzheimer's risk prediction | PEARL | 1000 Hz | 19 | 30s | 2 |

Table 10: Hyperparameters for T-EEG fine-tuning.

| Hyperparameters | Settings |
|---|---|
| Epochs | 50 |
| Batch size | 64 |
| Dropout | 0.1 |
| Optimizer | AdamW |
| Learning rate | 1e-4 |
| Adam $\beta$ | (0.9, 0.999) |
| Adam $\epsilon$ | 1e-8 |
| Weight decay | 5e-2 |
| Scheduler | CosineAnnealingLR |
| Cosine cycle epochs | 50 |
| Minimal learning rate | 1e-6 |
| Clipping gradient norm | 1 |

W, N1, N2, N3, REM. We apply a low-pass filter with a cut-off frequency at 30 Hz, sampling rate: 200 Hz, and 30-second window sample (6000 data points) to Sleep-EDFx.

**Evaluation.** We adopt the proposed subject-wise cross-validation protocol. We split the total dataset into 5 folds with the same number of subjects. We perform 5 fine-tunings, each involving a different fold as a testing dataset, and the remaining 4 folds serve as the training and validation sets. We randomly select training and validation data from these 4 folds, with a val-train ratio of 1:9.

### A.1.4 TUAB

**Description & Preprocessing.** TUAB consists of 409,455 10-second samples of subjects annotated as normal or abnormal (2-label classification). We apply a band-pass filter from 0.1 to 75 Hz, a notch filter at 50 Hz, sampling rate: 200 Hz, and 10-second window sample (2000 data points).

**Evaluation.** As TUAB has its own evaluation set, which we consider as the test set. We adopt the proposed cross-validation protocol for validation sets. We split all subjects into 4 folds of subjects. We then conduct 4 fine-tunings, each involving one fold of subjects as a validation set, and the remaining subjects serve as the training set. Generally, the train-valid-test ratio is 6:2:2.

### A.1.5 A&MISP

**Description & Preprocessing.** A&MISP consists of 1,881 four-second samples from 30 subjects, each annotated with one of four motor-imagery labels (4-class classification). We apply a band-pass filter from 1 to 50 Hz, a 50 Hz notch filter, re-referencing, per-channel standardization, ICA, and resampling to 200 Hz. Each sample is a 4-second window (800 data points).

**Evaluation.** We adopt a 5-fold cross-subject validation protocol stratified by gender using the available metadata. The samples from 30 patients are partitioned into five folds so that each fold

preserves the male–female ratio. We then conduct 5 fine-tunings, each involving one fold of subjects as a validation set, and the remaining subjects serve as the training set.

### A.1.6   N-FM

**Description & Preprocessing.**   N-FM consists of EEG samples recorded at 512 Hz in a character-recognition experiment, with each sample annotated with one of 94 character classes (94-class classification). We first select the Fq1 channel, then apply a band-pass filter from 1 to 50 Hz, a 50 Hz notch filter, re-referencing, per-channel standardization, and resample the data to 200 Hz.

**Evaluation.**   We adopt a 5-fold cross-class validation protocol over all 94 character classes, jointly using both male and female recordings. For each class, we partition them into five folds so that each fold contains approximately the same number of samples for that class, thereby preserving class balance across folds and gender. We perform 5 fine-tunings, each involving one fold as a validation set, and the remaining serve as the training set.

### A.1.7   EEGET-ALS

**Description & Preprocessing.**   EEGET-ALS contains EEG recordings from six ALS patients and 170 healthy controls, with 32 channels sampled at 256 Hz across nine scenarios involving imagined/executed limb movements, spelling, and rest. In our experiment, we use four labels (lift left hand, lift right hand, lift leg, rest). We select 19 channels, apply channel-wise demeaning, a 0.3-50 Hz band-pass filter, a 50 Hz notch filter, 4-second windows, resample to 200 Hz (800 data points), and perform per-channel normalization.

**Evaluation.**   We adopt a cross-population evaluation protocol that trains on healthy participants and tests on ALS patients. All healthy subjects are randomly split subject-wise into 85% training and 15% validation sets, while all ALS subjects are held out exclusively for testing. We perform 5 fine-tunings on data from the healthy training subjects, and use validation dataset used for model selection.

### A.1.8   M-EEG-EPI (EEG + BBB / EEG + TEXT)

**Description & Preprocessing.**   M-EEG-EPI comprises two modalities-EEG signals and BBB features-from 168 subjects performing an epilepsy detection task (2-label classification). For EEG, we apply a 0.3-75 Hz band-pass filter, a 50 Hz notch filter, resample to 200 Hz, and extract 10-second windows. For blood-based biomarker features, we apply z-score normalization. Each EEG window is then complemented with a vector of biomarker features.

In the EEG+text configuration, we use a subset of 158 subjects for epilepsy detection. For EEG, we apply the same preprocessing pipeline as above. Each EEG segment is paired with a same-day non-contrast brain MRI report. For the text modality, we select each subject's MRI report and encode it using the Clinical-T5 model from Google.

**Evaluation.**   For both configurations, we adopt a subject-wise 5-fold cross-validation protocol. The available subjects (168 for EEG + BBB and 158 for EEG + text) are splited into 5 folds with (approximately) the same number of subjects. We perform 5 fine-tunings, each involves a different fold as the test set, while the remaining 4 folds serve as the pool for training and validation. From these 4 folds, we randomly select training and validation data with a validation-to-training ratio of 2:8.

### A.1.9   M-EEG-TIA

**Description & Preprocessing.**   M-EEG-TIA comprises two modalities- EEG signals and BBB features- from 30 subjects for transient ischemic attack (TIA) detection (2-label classification). As in M-EEG-EPI, for EEG, we apply a 0.3-75 Hz band-pass filter, a 50 Hz notch filter, resample to 200 Hz, and extract 10-second windows (2,000 data points). For blood-based biomarker features, we apply z-score normalization. Each EEG window is then complemented with a vector of biomarker features.

**Evaluation.**   We follow the same subject-wise 5-fold cross-validation protocol as for M-EEG-EPI. For each run, one fold is held out as the test set, while the remaining 4 folds form the pool for

Table 11: Comparison of EEGPT with linear mapping to the 19 standard channels (w/ map) versus without linear mapping (w/o map).

| Tasks | Architectures | | Balanced Accuracy ↑ | | Cohen's Kappa / AUPR ↑ | | Weighted F1 / AUROC ↑ | |
|---|---|---|---|---|---|---|---|---|
| | | | Performance | Diff. | Performance | Diff. | Performance | Diff. |
| TUAB | EEGPT | w/ map | 0.8018 | | 0.8808 | | 0.8826 | |
| | | w/o map | 0.8136 | +1.47% | 0.8946 | +1.57% | 0.8916 | +1.02% |
| Sleep-EDFx | EEGPT | w/ map | 0.6585 | | 0.5963 | | 0.6976 | |
| | | w/o map | 0.6009 | -8.75% | 0.5556 | -6.83% | 0.6574 | -5.76% |

training and validation. We randomly split windows from these 4 folds into training and validation sets using a 2:8 validation-to-training ratio.

### A.1.10 M-EEG-PD

**Description & Preprocessing (EEG + BBB, PD).** M-EEG-PD is a multimodal downstream dataset extracted from M-EEG, containing two modalities- EEG signal and BBB features- for Parkinson's disease diagnosis (2-label classification). As in M-EEG-EPI and M-EEG-TIA, for EEG, we apply a 0.3-75 Hz band-pass filter, a 50 Hz notch filter, resample to 200 Hz, and extract 10-second windows (2,000 data points). For blood-based biomarker features, we apply z-score normalization. Each EEG window is then complemented with a vector of biomarker features.

**Evaluation.** We adopt the proposed subject-wise cross-validation protocol. We split the total dataset into 3 folds with the same number of subjects. We perform 3 fine-tunings, each involving a different fold as a testing dataset, and the remaining 2 folds serve as the training sets.

### A.1.11 ABLATION STUDY WITH LINEAR MAPPING ON EEGPT

We conducted additional experiments with EEGPT in which all datasets were fed in their native channel configuration, without any mapping to 19 channels. We used two datasets: Sleep-EDFx (2 channels) and TUAB (23 channels). For Sleep-EDFx, signals were passed directly to the encoder and use existing channels embeddings; for TUAB, we added 4 extra channel embeddings.

The results in the table 11, indicate that the impact of linear mapping is minimal. For Sleep-EDFx, the performance with linear mapping is slightly better than without it; for TUAB, the performance drop is marginal (approximately 1%).

## A.2 DETAILS ON MULTIMODAL FUSION FINETUNING

**Motivation.** We draw motivation from medical studies indicating that cognitive impairments, such as Alzheimer's disease, are often accompanied by measurable alterations in peripheral blood counts, reflecting changes in both the numbers and proportions of circulating cells (Shad et al., 2013; Zhang et al., 2022; Dzianok & Kublik, 2024). Importantly, blood-based biomarkers provide a low-cost and minimally invasive means of capturing such physiological signals. Inspired by this, we propose a multimodal pipeline that integrates blood test results with EEG data to facilitate earlier detection of cognitive decline and support timely clinical intervention.

**Multimodal fusion finetuning.** Formally, let $\boldsymbol{r} \in \mathbb{R}^m$ denote the normalized vector of blood-based biomarkers. We apply a lightweight projection network $\mathrm{MLP}(\cdot)$ that maps $\boldsymbol{r}$ into the EEG token embedding space:

$$\boldsymbol{q} = \mathrm{MLP}(\boldsymbol{r}) \in \mathbb{R}^d. \tag{1}$$

Given EEG embedded tokens $\boldsymbol{Z} = \mathcal{E}_\theta(\boldsymbol{X}) \in \mathbb{R}^{L \times d}$, we implement late fusion by treating $\boldsymbol{q}$ as a query attending to the EEG tokens:

$$\alpha = \mathrm{softmax}\left(\frac{(\boldsymbol{q}W_Q)(\boldsymbol{Z}W_K)^\top}{\sqrt{d_k}}\right), \qquad \boldsymbol{h} = \alpha(\boldsymbol{Z}W_V)W_O \in \mathbb{R}^d. \tag{2}$$

The resulting cross-modal representation $\boldsymbol{h}$ serves as input to a prediction head for downstream tasks. At a high level, we adopt cross-attention since it enables *adaptive alignment* between biomarker information and EEG dynamics: the biomarker query can selectively attend to the most informative EEG patterns rather than relying on a static combination. This flexibility is particularly important when the contribution of blood-based signals varies across patients or conditions.

Table 12: Alzheimer's risk prediction on the PEARL dataset. We compare unimodal EEG (pretrained using P-EEG) with multimodal EEG plus blood-based biomarkers (Concat. and Attention). Metrics are balanced accuracy, PR-AUC and ROC-AUC. Relative improvements (%) over EEG-only are shown in the **Gain** columns, with teal denoting improvements and magenta for drops.

| Task | Architecture | Metric | EEG-Only | | BBB-Only | | EEG + BBB (Concat.) | | EEG + BBB (Attention) | |
|------|-------------|--------|------|------|------|------|------|------|------|------|
| | | | Perf. | Gain | Perf. | Gain | Perf. | Gain | Perf. | Gain |
| **PEARL-MSIT** | **CBraMOD** | Balanced Accuracy | 0.5283 | | 0.543 | | 0.5515 | +4.39% | 0.6743 | +27.64% |
| | | AUPR | 0.5523 | | 0.526 | | 0.5609 | +1.56% | 0.7588 | +37.39% |
| | | AUROC | 0.5877 | | 0.603 | | 0.6148 | +4.61% | 0.7779 | +32.36% |
| | **EEGPT** | Balanced Accuracy | 0.4615 | | 0.543 | | 0.5505 | +19.29% | 0.5660 | +22.64% |
| | | AUPR | 0.4285 | | 0.526 | | 0.5319 | +24.13% | 0.5789 | +35.10% |
| | | AUROC | 0.4063 | | 0.603 | | 0.4974 | +22.42% | 0.5191 | +27.76% |
| **PEARL-SMT** | **CBraMOD** | Balanced Accuracy | 0.5296 | | 0.543 | | 0.5492 | +3.70% | 0.6213 | +17.32% |
| | | AUPR | 0.4692 | | 0.526 | | 0.6213 | +32.42% | 0.6773 | +44.35% |
| | | AUROC | 0.5040 | | 0.603 | | 0.6274 | +24.48% | 0.7156 | +41.98% |
| | **EEGPT** | Balanced Accuracy | 0.4746 | | 0.543 | | 0.4861 | +2.42% | 0.5627 | +18.56% |
| | | AUPR | 0.4132 | | 0.526 | | 0.5375 | +30.08% | 0.6109 | +47.85% |
| | | AUROC | 0.4222 | | 0.603 | | 0.4855 | +14.99% | 0.5651 | +33.85% |
| **PEARL-RST** | **CBraMOD** | Balanced Accuracy | 0.4375 | | 0.543 | | 0.6472 | +47.93% | 0.6960 | +59.09% |
| | | AUPR | 0.4445 | | 0.526 | | 0.7095 | +59.62% | 0.7772 | +74.85% |
| | | AUROC | 0.4580 | | 0.603 | | 0.6839 | +49.32% | 0.7783 | +69.93% |
| | **EEGPT** | Balanced Accuracy | 0.4366 | | 0.543 | | 0.4776 | +9.39% | 0.5753 | +31.77% |
| | | AUPR | 0.3925 | | 0.526 | | 0.4127 | +5.15% | 0.5985 | +52.48% |
| | | AUROC | 0.3949 | | 0.603 | | 0.4165 | +5.47% | 0.5483 | +38.85% |

## A.3 MORE RESULTS ON ALZHEIMER'S RISK PREDICTION ON THE PEARL DATASET

In this section, we report additional results on Alzheimer's risk prediction using the PEARL dataset. Specifically, we investigate the contribution of blood biomarkers when combined with EEG representations extracted from two foundation models (**CBraMod** and **EEGPT**). The goal is to assess (i) whether multimodal fusion with blood improves over EEG-only baselines, and (ii) how EEG compares to blood-only models in terms of predictive power.

In the PEARL dataset, the BBB includes: leukocytes (white blood cell count), erythrocytes (red blood cell count), hemoglobin, hematocrit, mean corpuscular volume (MCV), mean corpuscular hemoglobin (MCH), mean corpuscular hemoglobin concentration (MCHC), red cell distribution width (RDW-CV), platelet count, platelet distribution width (PDW), mean platelet volume (MPV), platelet large cell ratio (P-LCR), absolute counts of neutrophils, lymphocytes, monocytes, eosinophils, and basophils, as well as their relative percentages (neutrophils%, lymphocytes%, monocytes%, eosinophils%, basophils%), together with a standard lipid panel comprising total cholesterol, HDL-cholesterol, non-HDL cholesterol, LDL-cholesterol, and triglycerides.

In addition to evaluating the original checkpoints of **EEGPT** and **CBraMod**, we also pretrained both foundation models on our dataset and repeated the same experiments. This allows us to assess whether the observed multimodal gains are consistent across both the original and domain-adapted versions of the foundation models.

Table 12 reports results obtained with our domain-adapted checkpoints. We compare EEG-only and Blood-only models with multimodal EEG+Blood models (Concat and Attention fusion). Across both **CBraMod** and **EEGPT**, attention-based fusion consistently achieves the best performance, indicating that selective modality weighting is more effective than simple concatenation. In this setting, EEG-only models generally outperform Blood-only models, but combining EEG with blood further improves performance, confirming that blood biomarkers provide complementary information for Alzheimer's risk prediction when integrated with EEG signals.

Table 13 presents the corresponding results for the original (with less clinical information) checkpoints. Here, Blood-only models consistently outperform EEG-only models, and attention-based fusion again yields the strongest gains among multimodal strategies. The fact that multimodal EEG+Blood models improve over both unimodal baselines in both tables confirms that the benefit of incorporating blood biomarkers is robust.

Table 13: Alzheimer's risk prediction on the PEARL dataset. We compare unimodal EEG (using the original checkpoints) with multimodal EEG plus blood-based biomarkers (Concat. and Attention). Metrics are balanced accuracy, PR-AUC and ROC-AUC. Relative improvements (%) over EEG-only are shown in the **Gain** columns, with teal denoting improvements and magenta for drops.

| Task | Architecture | Metric | EEG-Only | | BBB-Only | | EEG + BBB (Concat.) | | EEG + BBB (Attention) | |
|------|--------------|--------|----------|------|----------|------|---------------------|------|-----------------------|------|
| | | | Perf. | Gain | Perf. | Gain | Perf. | Gain | Perf. | Gain |
| PEARL-MSIT | CBraMOD | Balanced Accuracy | 0.4816 | | 0.543 | | 0.5263 | +9.28% | 0.6373 | +32.33% |
| | | AUPR | 0.5597 | | 0.526 | | 0.6013 | +7.43% | 0.6863 | +22.62% |
| | | AUROC | 0.5818 | | 0.603 | | 0.5979 | +2.77% | 0.7235 | +24.36% |
| | EEGPT | Balanced Accuracy | 0.4550 | | 0.543 | | 0.4968 | +9.19% | 0.5560 | +22.20% |
| | | AUPR | 0.4840 | | 0.526 | | 0.5767 | +19.15% | 0.6056 | +25.12% |
| | | AUROC | 0.4035 | | 0.603 | | 0.4915 | +21.81% | 0.5023 | +24.49% |
| PEARL-SMT | CBraMOD | Balanced Accuracy | 0.5280 | | 0.543 | | 0.4982 | -5.64% | 0.6288 | +19.09% |
| | | AUPR | 0.4661 | | 0.526 | | 0.5656 | +21.35% | 0.6043 | +29.65% |
| | | AUROC | 0.4985 | | 0.603 | | 0.5946 | +19.28% | 0.6554 | +31.47% |
| | EEGPT | Balanced Accuracy | 0.4312 | | 0.543 | | 0.4310 | -0.05% | 0.5226 | +21.20% |
| | | AUPR | 0.3982 | | 0.526 | | 0.4462 | +12.05% | 0.5745 | +44.27% |
| | | AUROC | 0.3805 | | 0.603 | | 0.4072 | +7.02% | 0.5285 | +38.90% |
| PEARL-RST | CBraMOD | Balanced Accuracy | 0.4504 | | 0.543 | | 0.5606 | +24.47% | 0.5793 | +28.62% |
| | | AUPR | 0.3927 | | 0.526 | | 0.6600 | +68.07% | 0.6666 | +69.75% |
| | | AUROC | 0.3997 | | 0.603 | | 0.6098 | +52.56% | 0.6416 | +60.52% |
| | EEGPT | Balanced Accuracy | 0.3952 | | 0.543 | | 0.4096 | +3.64% | 0.5722 | +44.79% |
| | | AUPR | 0.3556 | | 0.526 | | 0.3910 | +9.96% | 0.4856 | +36.56% |
| | | AUROC | 0.3281 | | 0.603 | | 0.3742 | +14.05% | 0.4310 | +31.36% |

## A.4 More Details on Neurological Disorders Prediction

**Lab values panel.** In the M-EEG cohort, the BBB vector is constructed routine blood tests. Specifically, it includes absolute and relative counts of basophils, eosinophils, lymphocytes, monocytes, and neutrophils; hemoglobin, platelet count, red blood cell count, white blood cell count, hematocrit, mean corpuscular volume (MCV), mean corpuscular hemoglobin (MCH), mean corpuscular hemoglobin concentration (MCHC), red cell distribution width (RDW), and mean platelet volume (MPV); serum electrolytes, including sodium (Na+), potassium (K+), and chloride (Cl-); liver enzymes alanine aminotransferase (ALT/GPT), aspartate aminotransferase (AST/GOT), and gamma-glutamyl transferase (GGT); renal and nitrogen-metabolism markers (serum creatinine, blood urea); uric acid; total calcium; a lipid profile comprising total cholesterol, high-density lipoprotein cholesterol (HDL-C), low-density lipoprotein cholesterol (LDL-C), and triglycerides; as well as glucose and glycated hemoglobin (HbA1c) as markers of short- and long-term glycemic status.

**Ablation study on the impact of free-text clinical notes.** We further demonstrate the value of the added text modality. In our setting, the text corresponds to free-text clinical notes that summarize MRI findings for each patient, for example, "Chronic small-vessel white-matter changes in the periventricular region and bilateral centrum semiovale. Right maxillary sinus retention cyst". We adopt the same late-fusion finetuning strategy as for the blood modality. Specifically, each text sentence is fed into a T5 encoder, whose outputs are used as query vectors to attend to the EEG encoder representations. As shown in Table 14, without textual information, the models perform only slightly better than random guessing; once text is incorporated, their performance improves substantially, with CBraMOD gaining 17.66% and EEGPT gaining 31.97% in balanced accuracy.

Table 14: Ablation study for Neurological disorders prediction on the M-EEG dataset. We compare unimodal EEG (Base) multimodal EEG plus free-text clinical notes (w/ Text), with teal denotes the relative improvements over the EEG-only baseline.

| Tasks | Architectures | | Balanced Accuracy | | AUPR | | AUROC | |
|-------|---------------|--|-------------------|------|------|------|-------|------|
| | | | Performance | Gain | Performance | Gain | Performance | Gain |
| Epilepsy | CBraMOD | Base | 0.5282 | | 0.4317 | | 0.5550 | |
| | | w/ Text | 0.6215 | +17.66% | 0.5846 | +35.42% | 0.6233 | +12.31% |
| | EEGPT | Base | 0.5120 | | 0.4058 | | 0.5056 | |
| | | w/ Text | 0.6757 | +31.97% | 0.6783 | +67.15% | 0.7194 | +42.29% |

## A.5 ANALYSIS OF DISTRIBUTIONAL DIFFERENCES ACROSS INSTITUTIONS

To address potential sampling biases, we analyzed the data characteristics from the two participating institutions. However, a direct comparison of patient demographics was not feasible. Due to differing data collection and privacy protocols, demographic information (age, gender) was not available for Hospital A and was only partially available (2,185 of 5,134 subjects having age label, 5,104 of 5,134 subjects having gender label) for Hospital B.

Our analysis therefore focuses on (1) reporting the available demographic subset from Hospital B, and (2) quantifying the clear inter-institutional differences in recording statistics and equipment configurations.

### A.5.1 AVAILABLE PATIENT DEMOGRAPHICS (HOSPITAL B)

As stated, demographic data for Hospital A was unavailable. We report the statistics for the available subset of Hospital B in Table 2 and Figure 2. Due to this limitation, a direct statistical comparison of demographics between sites could not be performed.

Based on the available records from Hospital B, the age-labeled subset ($N = 2,185$) ranges from 1 to 104 years, with a median age of 46. Regarding gender ($N = 5,104$), the distribution is imbalanced: female patients constitute the majority (3,748 subjects; 73.0%), compared to 1,356 male subjects (26.4%).

### A.5.2 COMPARISON OF RECORDING STATISTICS AND EQUIPMENT BIAS

While demographics could not be directly compared, our analysis of recording data and equipment configurations revealed significant inter-institutional differences.

**Recording Statistics:** We analyzed the yearly and duration distributions for both sites.

- **For Hospital A,** the distributions are shown in Figure 4.
- **For Hospital B,** the distributions are shown in Figure 5.

Visually comparing the two, we observe distinct temporal patterns: Hospital A contributed the majority of its recordings during 2021–2022, whereas Hospital B's contributions are concentrated in the more recent 2024–2025 period. This complementary distribution enhances the temporal diversity of the M-EEG dataset. Regarding recording duration, we observe notable differences between the sites:

- **Hospital A:** The recordings have a mean duration of 1,043.54 seconds, with the longest record lasting 7,975 seconds. The majority of recordings (923 of 947) fall within the range of 0 to 2,000 seconds.
- **Hospital B:** The recordings are generally shorter, with a mean duration of 163.43 seconds. However, this site includes significant outliers, with the longest record lasting 48,802 seconds. Similar to Hospital A, the vast majority of records (5,204 of 5,272) have a duration under 2,000 seconds.

**Equipment Bias:** The most pronounced difference is the equipment bias, which we explicitly quantify in Table 3. The institutions used entirely different hardware, resulting in a significant domain shift in sampling rate (200 Hz vs. 500 Hz) and channel count (22 vs. 44). However, this heterogeneity enhances the ecological validity of the dataset. It mirrors the reality of multi-center clinical data, providing a challenging testbed for developing models that are robust to hardware variations.

### A.6 DESCRIPTION OF THE BIDS STRUCTURE OF THE DATABASE

In this study, we organized our database following the **Brain Imaging Data Structure (BIDS) specification**, version 1.8.0. BIDS is a community-driven standard that provides a uniform way to arrange neuroimaging and physiological datasets, ensuring consistency, interoperability, and reproducibility across studies.

By adopting BIDS v1.8.0, we gain several advantages:

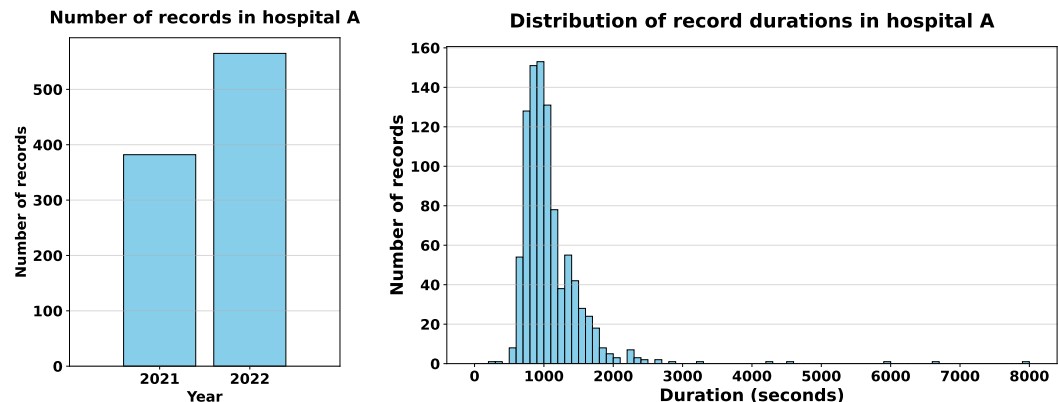

Figure 4: Yearly and duration distribution of subjects' recordings collected from Hospital A in M-EEG dataset

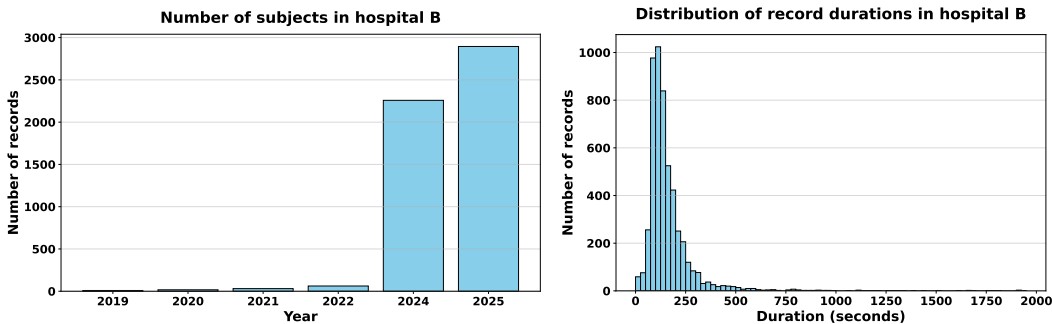

Figure 5: Yearly and duration distribution of subjects' recordings collected from Hospital B in M-EEG dataset

- **Standardization**: Data from different acquisition sites and modalities (e.g., EEG signals, clinical laboratory results) are represented in a consistent format, reducing ambiguity in interpretation.

- **Compatibility**: The dataset can be directly integrated with existing BIDS-aware software tools for preprocessing, quality control, and statistical analysis.

- **Reproducibility**: Researchers can reuse the dataset with minimal manual curation, which facilitates replication studies and meta-analyses.

- **Extensibility**: Beyond EEG recordings, our design includes phenotype-level information (e.g., laboratory test results), enabling multimodal analysis that links neurophysiological data with clinical variables.

At the top level, the dataset is structured according to the BIDS hierarchy, which includes:

- `dataset_description.json`: Contains metadata describing the dataset, its authorship, and BIDS compliance.

- `participants.tsv` and `participants.json`: Contain participant-level demographic and group information.

- `phenotype/`: Contains clinical laboratory test results in `results.tsv` and related metadata in `results.json`.

- `sub-xxxx/`: Contain subject-specific data, including an `eeg/` subfolder with EEG recordings, associated metadata, channel information, and a `sub-xxxx_scans.tsv` file documenting recording timestamps.

This organization ensures that the dataset is self-describing and can be recognized by BIDS-compatible tools without requiring additional documentation.

## A.7 EXTENDED RELATED WORK

This section positions our work within the broader literature on multimodal EEG benchmarks and standardization. A comprehensive comparison of current state-of-the-art and ours is summarized in Table 15.

**Benchmarks for EEG and time series.** There are studies that have already standardized multiple datasets across regions, groups, and conditions Chevallier et al. (2024); Gagnon-Audet et al. (2023); Charest et al. (2025); Aristimunha et al. (2025); Darvishi-Bayazi et al. (2024); Ferrante et al. (2024), but their objectives and scopes differ substantially from ours. Our work is the first to standardize *multimodal* EEG-based clinical datasets for benchmarking foundation models across diverse EEG-related tasks. We create a unified and standardized framework in which each sample may include EEG signals alongside zero, one, or multiple clinical modalities (e.g., laboratory test results), enabling benchmarking across a broad range of EEG-related downstream tasks under a consistent evaluation protocol. For the multimodal datasets in particular, our benchmarking effort focuses on *neurological disease diagnosis*, a clinically meaningful and technically challenging setting. Among prior works, only Chevallier et al. (2024) and Gagnon-Audet et al. (2023) qualify as benchmark efforts: Chevallier et al. (2024) focuses on BCI reproducibility using single-modality EEG for BCI control, while Gagnon-Audet et al. (2023) is a cross-domain generalization benchmark across heterogeneous time series where EEG appears only as two datasets and the goal is to benchmark domain generalization methods. Thus, neither the dataset scope nor the benchmarking objectives overlap with ours.

**Multimodal neuroimaging, physiological signals, and cross-domain EEG.** We contribute the first multimodal EEG clinical dataset collected from two hospitals outside the US. Our dataset includes paired EEG + laboratory test data, enabling multimodal learning for neurological disease tasks. None of the prior works include such multimodality. While Charest et al. (2025) and Ferrante et al. (2024) include EEG/MEG or EEG/fMRI, these modalities come from separate datasets and are not aligned within the same sample. In contrast, each sample in our dataset contains multiple synchronized clinical modalities, enabling models to learn richer physiological relationships that have not been explored in previous benchmarks. The EEG Foundation Challenge Aristimunha et al. (2025) constructs a large-scale cohort of EEG recordings with demographic information and studies cross-task and cross-subject decoding, including zero-shot cross-domain generalization, but it is still built around a single dataset and remains essentially unimodal at the signal level. Darvishi-Bayazi et al. (2024) studies cross-dataset transfer learning for pathology detection using TUAB and NMT scalp EEG, but the setting is strictly unimodal (EEG only) and framed as transfer between two datasets rather than as a general benchmark for EEG foundation models. The Brant series Zhang et al. (2023); Yuan et al. (2024); Zhang et al. (2024) further develops foundation models for intracranial and scalp brain signals and a unified alignment framework between EEG and other physiological signals (EOG, ECG, EMG). Brant Zhang et al. (2023) scales foundation models to intracranial SEEG by pretraining exclusively on a large private SEEG cohort, targeting invasive neural recordings rather than scalp EEG. Brant-2 Yuan et al. (2024) extends this line of work by training a unified backbone on both SEEG and EEG (private SEEG + TUEG), but still operates within a single-modality neural signal space and does not explore explicit multimodal alignment. Brant-X Zhang et al. (2024) moves toward multimodality by jointly modeling EEG with other physiological signals (EOG, ECG, EMG) on CAP, ISRUC, and HMC, focusing on cross-signal alignment between biosignals rather than fusion multiple modalities.

**Positioning and novelty of our benchmark.** Beyond benchmarking, we propose and validate a new multimodal EEG model showing significant performance gains for Alzheimer's disease prediction. Our multimodal fusion model integrates EEG with additional clinical modalities, and our experiments show that adding complementary modalities yields substantial improvements in Alzheimer's prediction accuracy, demonstrating the scientific value of multimodal EEG integration. While prior works address unimodal EEG, cross-modal reconstruction (e.g., EEG→fMRI), unimodal transfer learning, or foundation models and alignment frameworks for brain and physiological signals, none of them provide multimodal clinical data, a unified benchmark specifically designed for EEG foundation models, or evidence that multimodality improves disease prediction. In summary, the key

added values of our benchmark are: (i) a clinically oriented, multimodal EEG benchmark not present in prior studies; (ii) a new dataset from two non-US hospitals with paired EEG + lab results per sample; and (iii) a novel multimodal EEG model validated through extensive experiments.

Table 15: Comparison of our multimodal benchmark and standardization pipeline with prior works.

| References | Modalities of Each Sample | Datasets | Tasks |
|---|---|---|---|
| Chevallier et al. (2024) | Only EEG | 36 publicly available datasets, including motor imagery (14), P300 (15), and SSVEP (7) | Benchmark for BCI reproducibility |
| Gagnon-Audet et al. (2023) | One type of time series | CAP, SEDFx | Benchmark for out-of-distribution generalization |
| Charest et al. (2025) | Either EEG or fMRI | Natural Scenes (7T fMRI responses), NSD-EEG (EEG) | EEG-to-fMRI generation |
| Aristimunha et al. (2025) | EEG and demographic information | 1 Dataset: EEG signals (128 channels) recorded from over 3,000 child to young adult | Zero-shot cross-domain generalization |
| Darvishi-Bayazi et al. (2024) | EEG | Temple University Hospital Abnormal (TUAB), and NUST-MH-TUKL (NMT) scalp EEG | Pathology classification task |
| Ferrante et al. (2024) | Either EEG, MEG, or fMRI | ImageNetEEG dataset, THINGS-MEG dataset, Natural Scenes Dataset (NSD) | Multimodal alignment |
| Zhang et al. (2023) | Only SEEG | a private SEEG dataset | Towards foundation models for intracranial neural signal |
| Yuan et al. (2024) | either SEEG or EEG | a private SEEG dataset, TUEG | Towards foundation models for brain signals |
| Zhang et al. (2024) | either EEG, EOG, ECG, or EMG | CAP, ISRUC, and HMC | Multimodal alignment |
| **Ours** | EEG, lab values and clinical notes | M-EEG, T-EEG, TUEG, NMT Scalp | Multimodal EEG fusion benchmark |

## A.8 LIMITATIONS

The robustness gains from incorporating regional data are marginal but consistent, indicating steady benefits even at limited scale. These results provide encouraging evidence that regional coverage can enhance generalization, though M-EEG remains smaller than corpora such as TUEG or HEEDB. As we expand data collection to achieve greater balance, future work will more fully explore the role of regional diversity in building robust EEG foundation models.

