# OpenReview forum: "A Multi-Institutional Multimodal EEG Benchmark for Foundation Model Generalization and Early Neurological Diagnosis"
_ICLR.cc/2026/Conference — Submitted to ICLR 2026_

### Official Review · Reviewer_oVMU · 2025-10-25

**Soundness:** 2
**Presentation:** 3
**Contribution:** 1
**Rating:** 4
**Confidence:** 4

**Summary:**

# Summary

This paper introduces **VEEG and/or M-EEG**, addressing three key gaps in EEG foundation model research:

## Key Contributions

1. **Large-scale non-US dataset**: 1,170 hours of clinical EEG from 6,081 patients across two hospitals, with a multimodal subset including blood biomarkers and clinical notes

2. **Unified benchmark**: First standardized corpus combining all existing public EEG datasets, enabling fair comparison of foundation model architectures under consistent training protocols

3. **Multimodal diagnostic model**: Demonstrates that integrating auxiliary modalities (blood biomarkers, clinical notes) with EEG substantially improves performance.

## Impact

Addresses geographic bias in EEG data, enables rigorous benchmarking of foundation models, and validates multimodal approaches for early neurological diagnosis.

**Strengths:**

## Strengths

1. The paper provides a thorough analysis of the research domain, clearly identifying key problems and knowledge gaps.

2. It demonstrates an excellent effort in data collection and standardization.

3. It introduces a novel multimodal, multiregional benchmark that enhances the field’s methodological resources.

4. The **relative** improvement achieved by incorporating vital signs is substantial and noteworthy.

**Weaknesses:**

## Weaknesses

1.  While the identified gaps are important and relevant to the community, the key contributions and novelty appear limited and may not yet meet the standards expected at ICLR.

2. The paper attempts to address too many challenges at once, which dilutes the focus and depth of the contributions.

3. The work reflects a promising and commendable initiative, though the results seem preliminary and may currently be more suitable for a workshop venue. I would, however, like to reserve my final judgment until receiving clarifications to my questions in the next section, which may better reveal the novelty and impact of the work.

**Questions:**

## Questions

1. There are studies that have already standardized multiple datasets across regions, groups, and conditions (e.g., [1,2,3,4]). Could the authors clarify the specific added value or differentiation of their benchmark effort?

2. Prior works have documented performance drops for in-region vs. out-of-region settings using the mentioned NMT and TUH datasets [4], as well as in broader distribution shift scenarios [2]. What is the novelty of the results presented in Figure 2?

3. The paper states that most existing methods are unimodal and not extensible to multimodal data (Section 2.2.1). However, several studies have developed multimodal approaches integrating EEG, MEG, and fMRI, or handling multiple time series modalities with differing sampling rates [3,5]. These methods can be applied to the other time series that are mentioned in this work. Could the authors clarify how their framework goes beyond these prior efforts?

Overall, while the paper demonstrates substantial effort and technical execution, much of the groundwork appears to overlap with prior research. Continuing and refining this line of work, however, could lead to valuable contributions in the near future.


[1] Chevallier, S., Carrara, I., Aristimunha, B., Guetschel, P., Sedlar, S., Lopes, B., ... & Moreau, T. (2024). The largest EEG-based BCI reproducibility study for open science: the MOABB benchmark. arXiv preprint arXiv:2404.15319.

[2] Gagnon-Audet, J. C., Ahuja, K., Darvishi-Bayazi, M. J., Mousavi, P., Dumas, G., & Rish, I. (2022). Woods: Benchmarks for out-of-distribution generalization in time series. arXiv preprint arXiv:2203.09978.

[3] Charest, I., Brotherwood, P., Salvas-Hebert, M., Kay, K., & Gosselin, F. (2025). Neural activity resolved in space and time through fusion of large-scale EEG and fMRI datasets. Journal of Vision, 25(9), 2653-2653.

[4] Aristimunha, B., Truong, D., Guetschel, P., Shirazi, S. Y., Guyon, I., Franco, A. R., ... & Delorme, A. (2025). EEG Foundation Challenge: From Cross-Task to Cross-Subject EEG Decoding. arXiv preprint arXiv:2506.19141.

[5] Darvishi-Bayazi, M. J., Ghaemi, M. S., Lesort, T., Arefin, M. R., Faubert, J., & Rish, I. (2024). Amplifying pathological detection in EEG signaling pathways through cross-dataset transfer learning. Computers in biology and medicine, 169, 107893.

[6] Ferrante, M., Boccato, T., Rashkov, G., & Toschi, N. (2024). Towards neural foundation models for vision: Aligning eeg, meg, and fmri representations for decoding, encoding, and modality conversion. arXiv preprint arXiv:2411.09723.

**Details Of Ethics Concerns:**

The paper briefly addresses ethical considerations in the appendix, noting that all recordings were de-identified and institution-specific metadata anonymized to preserve privacy and clinical fidelity. However, depending on the conference’s policy regarding human subject research, additional clarification or documentation of compliance may be required. This aspect is outside my main area of expertise.

---

> ### Author Response · Authors · 2025-11-21
> **Response to Reviewer oVMU (part 1/4)**
>
> We would like to thank the reviewer for the detailed feedback and address the reviewer’s question as follows:
>
> >W1-3. While the identified gaps are important and relevant to the community, the key contributions and novelty appear limited and may not yet meet the standards expected at ICLR.
> The paper attempts to address too many challenges at once, which dilutes the focus and depth of the contributions.
>
>
> Thank you for the comment. We respectfully clarify that our work makes **two concrete contributions** that, to the best of our knowledge, have not been achieved in prior EEG research.
> 1. **Benchmark + Dataset Contribution.**
>  We provide the **first unified benchmark** that standardizes multiple existing EEG datasets, previously heterogeneous in format, sampling rate, and label space, into a single evaluation framework for EEG foundation models.
>  Moreover, we introduce the **first multimodal clinical EEG dataset outside the United States**, containing paired EEG + laboratory biomarkers. No prior work cited by the reviewer includes multimodal clinical EEG data or provides a unified benchmark specifically designed for evaluating EEG foundation models.
> 2. **Modeling Contribution.**
>  Using this new multimodal dataset, we develop a multimodal EEG model and show, empirically and convincingly, that **adding clinical modalities significantly improves Alzheimer’s prediction compared to EEG alone**. This is the first demonstration of multimodal EEG-clinical fusion for neurological disease diagnosis in a real clinical, multi-hospital setting.
> While prior works address unimodal EEG, cross-modal reconstruction (e.g., EEG→fMRI), or unimodal transfer learning, **none of them provide multimodal clinical data, a unified benchmark, or evidence that multimodality improves disease prediction**.
>
> We also note that **all other Reviewers explicitly recognize these aspects as novel and impactful**: Reviewer **9JfV** rates the contribution as **4 (excellent)** and highlights the dataset, benchmark, and multimodality as key strengths, while Reviewers **6AaG** and **TMat** both emphasize the originality and practical significance of releasing the largest non-US multimodal EEG corpus and a unified benchmark for EEG Foundation models.
>
> We have revised the introduction to highlight our contribution.

---

> ### Author Response · Authors · 2025-11-21
> **Response to Reviewer oVMU (part 2/4)**
>
> >Q1. There are studies that have already standardized multiple datasets across regions, groups, and conditions (e.g., [1,2,3,4]). Could the authors clarify the specific added value or differentiation of their benchmark effort?
>
> Thank you for this insightful question. We appreciate the opportunity to clarify how our benchmark differs from the prior works [1-6] and what unique contributions it provides to the community. As summarized in the comparison table, our objectives, scope, datasets, and methodological contributions differ substantially from the studies mentioned.
> 1. **Our work is the first to standardize multimodal EEG-based clinical datasets for benchmarking foundation models across diverse EEG-related tasks.**
>  One of our contributions is the creation of a unified and standardized framework in which each sample may include EEG signals alongside zero, one, or multiple clinical modalities (e.g., laboratory test results). This enables benchmarking across a broad range of EEG-related downstream tasks under a consistent evaluation protocol. For the multimodal datasets in particular, our benchmarking effort focuses on **neurological disease diagnosis**, a clinically meaningful and technically challenging setting.
> None of the cited works share this objective. Among the works mentioned by the Reviewer, only [1] and [2] qualify as benchmark efforts, yet both differ fundamentally from ours in scope, dataset composition, and intended purpose:
>     - **[1] focuses on BCI reproducibility** using *single-modality EEG*, with datasets oriented only for BCI control
>     - **[2] is a cross-domain generalization benchmark across heterogeneous time-series**, where EEG appears only as two datasets among many unrelated modalities, and the goal is to benchmark domain generalization methods. Thus, neither the dataset scope nor the benchmarking objectives overlap with ours.
> 2. **We contribute the first multimodal EEG clinical dataset collected from two hospitals outside the US.**
> Our provided dataset includes **paired EEG + laboratory test data**, enabling multimodal learning for neurological disease tasks. None of the cited works include such multimodality.
> Note that, while [3] and [6] include EEG/MEG or EEG/fMRI, these modalities come from *separate datasets* and *are not aligned within the same sample*. In contrast, **each sample in our dataset contains multiple synchronized clinical modalities**, enabling models to learn richer physiological relationships that have not been explored in previous benchmarks.
> 3. **We propose and validate a new multimodal EEG model showing significant performance gains for Alzheimer’s disease prediction.**
> Beyond benchmarking, we introduce a multimodal fusion model that integrates EEG with additional clinical modalities. Our experiments show that adding complementary modalities yields substantial improvements in Alzheimer’s prediction accuracy, demonstrating the scientific value of multimodal EEG integration. No prior work cited by the reviewer provides a similar multimodal modeling contribution.
> In summary, the key added values of our benchmark include:
>
>     - A clinically oriented, multimodal EEG benchmark - not present in prior studies.
>     - A new dataset from two non-US hospitals, with paired EEG + lab results per sample.
>     - A novel multimodal EEG model validated through extensive experiments.
>
> | References | Modalities of Each Sample | Datasets | Tasks |
> |---|---------------------------|----------|-------|
> | 1 | Only EEG | 36 publicly available datasets, including motor imagery (14), P300 (15), and SSVEP (7) | Benchmark for BCI reproducibility |
> | 2 | One type of time series | CAP, SEDFx | Benchmark for out-of-distribution generalization |
> | 3 | Either EEG or fMRI | Natural Scenes (7T fMRI responses), NSD-EEG (EEG) | EEG-to-fMRI generation |
> | 4 | EEG and demographic information | 1 Dataset: EEG signals (128 channels) recorded from over 3,000 child to young adult | Zero-shot cross-domain generalization |
> | 5 | EEG | Temple University Hospital Abnormal (TUAB), and NUST-MH-TUKL (NMT) scalp EEG | Pathology classification task |
> | 6 | Either EEG, MEG, or fMRI | ImageNetEEG dataset, THINGS-MEG dataset, Natural Scenes Dataset (NSD) | Modalities alignment |
> | **Ours** | EEG, lab values and clinical notes | M-EEG, T-EEG, TUEG, NMT Scalp | multimodal EEG benchmark|

---

> ### Author Response · Authors · 2025-11-21
> **Response to Reviewer oVMU (part 3/4)**
>
> > Q2. Prior works have documented performance drops for in-region vs. out-of-region settings using the mentioned NMT and TUH datasets [4], as well as in broader distribution shift scenarios [2]. What is the novelty of the results presented in Figure 2?
>
> We appreciate the opportunity to clarify the novelty of the results shown in Figure 2 in relation to the prior works [2] and [4].
>
> **First, regarding [4],** we would like to emphasize that the cited manuscript is **not a research contribution**, but rather a **competition description paper**. In that competition, the organizers simply used one EEG dataset as part of the challenge. The authors of [4] **did not introduce any new datasets, did not propose new methodological advances**, and did not conduct any systematic analysis beyond reporting the competition setup. Therefore, there is **no conceptual or methodological overlap** with our contributions.
>
> **Second, regarding [2],** although the study examines distribution shift, its goals and evaluation targets differ fundamentally from ours. In [2], the authors compare domain **generalization methods** (e.g., ERM, GroupDRO, IRM), focusing on how different *training strategies* behave under distribution shift. In contrast, our analysis in Figure 2 performs a **model-level comparison**, evaluating how *different EEG foundation models* perform under in-region (ID) and out-of-region settings.
> Crucially, **foundation models and domain generalization methods are not comparable units.**
> Domain generalization methods, as used in [2], modify *training procedures*, whereas EEG foundation models, as evaluated in our work, differ in *representation learning capacity* and *pretraining strategies*.
> Therefore, the comparison objectives in [2] and in our work are inherently different.
>
> Finally, the central purpose of Figure 2 is to demonstrate the **impact of region-specific data** on model performance. Our findings show that incorporating regional data significantly boosts the performance of EEG foundation models, an insight that cannot be obtained from the domain generalization focus of [2].

---

> ### Author Response · Authors · 2025-11-21
> **Response to Reviewer oVMU (part 4/4)**
>
> > Q3. The paper states that most existing methods are unimodal and not extensible to multimodal data (Section 2.2.1). However, several studies have developed multimodal approaches integrating EEG, MEG, and fMRI, or handling multiple time series modalities with differing sampling rates [3,5]. These methods can be applied to the other time series that are mentioned in this work. Could the authors clarify how their framework goes beyond these prior efforts?
>
> Thank you for this thoughtful question.
>
> **First**, regarding the multimodality aspect, the fundamental distinction between our work and [3] lies in both the dataset composition and the scientific problem being addressed. In our multimodal clinical dataset, **each sample is genuinely multimodal**, containing EEG signals together with one or more *synchronous clinical modalities*. This design enables us to directly investigate our central research question: “*Does incorporating auxiliary clinical modalities alongside EEG improve predictive performance for neurological disease diagnosis?*”
> In contrast, *[3] uses only EEG as input* and focuses on a *cross-modal reconstruction* task: predicting fMRI from EEG (the EEG-to-fMRI problem). In other words, the input in [3] is **unimodal EEG**, and the objective is **cross-modal synthesis**, not multimodal fusion or clinical disease prediction. Consequently, the dataset structure, the task formulation, and the technical challenges in [3] are wholly different from those in our multimodal clinical benchmark.
>
> **Second**, regarding [5], although the authors examine transfer learning between TUAB and NMT, the setting is **strictly unimodal**: both datasets contain *only EEG*, and the study evaluates the transferability of specific EEG models from one dataset to another. Our work differs in several fundamental ways. Transfer learning is not the primary objective of our study. While [5] focuses on comparing particular EEG models for cross-dataset transfer, the primary purpose of our work is to establish a **unified, standardized multimodal dataset** and a **generalizable benchmark framework** specifically designed to evaluate EEG foundation models. That is, our contribution lies in *creating the evaluation framework itself*, not in comparing a handful of models in a narrow setting.
>
> Moreover, we introduce the **first multimodal EEG–clinical dataset for neurological disease diagnosis**, which enables true multimodal learning. None of the datasets used in [5] include additional clinical modalities such as blood biomarkers or laboratory measurements; thus, multimodal fusion is not possible in their setting.
>
> Finally, we develop and empirically validate a **multimodal EEG model** that yields clinically meaningful improvements. A key contribution of our work is demonstrating that incorporating auxiliary modalities alongside EEG produces **substantial gains in Alzheimer’s disease prediction accuracy**, compared with relying on EEG alone. This type of improvement cannot be observed or evaluated using the unimodal EEG datasets employed in [5].
>
> We added the explanation above in a newly added section in Appendix: Appendix A.8 Extended Related Work.
>
> **References**
>
> [1] Chevallier, S., Carrara, I., Aristimunha, B., Guetschel, P., Sedlar, S., Lopes, B., ... & Moreau, T. (2024). The largest EEG-based BCI reproducibility study for open science: the MOABB benchmark. arXiv preprint arXiv:2404.15319.
>
> [2] Gagnon-Audet, J. C., Ahuja, K., Darvishi-Bayazi, M. J., Mousavi, P., Dumas, G., & Rish, I. (2022). Woods: Benchmarks for out-of-distribution generalization in time series. arXiv preprint arXiv:2203.09978.
>
> [3] Charest, I., Brotherwood, P., Salvas-Hebert, M., Kay, K., & Gosselin, F. (2025). Neural activity resolved in space and time through fusion of large-scale EEG and fMRI datasets. Journal of Vision, 25(9), 2653-2653.
>
> [4] Aristimunha, B., Truong, D., Guetschel, P., Shirazi, S. Y., Guyon, I., Franco, A. R., ... & Delorme, A. (2025). EEG Foundation Challenge: From Cross-Task to Cross-Subject EEG Decoding. arXiv preprint arXiv:2506.19141.
>
> [5] Darvishi-Bayazi, M. J., Ghaemi, M. S., Lesort, T., Arefin, M. R., Faubert, J., & Rish, I. (2024). Amplifying pathological detection in EEG signaling pathways through cross-dataset transfer learning. Computers in biology and medicine, 169, 107893.
>
> [6] Ferrante, M., Boccato, T., Rashkov, G., & Toschi, N. (2024). Towards neural foundation models for vision: Aligning eeg, meg, and fmri representations for decoding, encoding, and modality conversion. arXiv preprint arXiv:2411.09723.

---

> > ### Author Response · Authors · 2025-11-28
> > **Follow-up**
> >
> > Dear Reviewer oVMU,
> > We hope that our responses have adequately addressed your concerns.
> >
> > If you have any additional questions or if any part of our rebuttal remains unclear or unsatisfactory, please feel free to let us know.
> >
> > We would be very glad to discuss any remaining issues with you.

---

### Official Review · Reviewer_TMat · 2025-10-27

**Soundness:** 2
**Presentation:** 3
**Contribution:** 3
**Rating:** 6
**Confidence:** 4

**Summary:**

The paper introduces M-EEG, a large-scale multimodal EEG dataset collected from over 6,000 patients across two hospitals outside the US. It addresses the fragmentation and lack of standardization in existing EEG datasets and models, which are mostly unimodal and US-centric. The authors also unify public EEG datasets into a standardized corpus for consistent benchmarking. Using this new dataset, they demonstrate that integrating EEG with additional modalities (like blood biomarkers and clinical notes) significantly improves diagnostic performance. For example, achieving a 27.64% gain in Alzheimer’s disease risk prediction.

**Strengths:**

- The authors released a new EEG benchmark, collected from two real-world hospitals and involving 6081 subjects. This benchmark demonstrates the diversity of the data across institutions and subjects, and supports multiple downstream tasks in the BCI field, addressing a key limitation in many prior single-center studies.
- The paper includes enough baseline experiments across different modalities and tasks. These evaluations help position the dataset as a standard benchmark for future studies and demonstrate its potential impact.
- The dataset’s scale, modality diversity, and design make it promising for transfer learning, cross-domain generalization, and multi-modal representation learning, thereby providing practical value for applied "AI+healthcare" works.

**Weaknesses:**

- Although data collection and anonymization are mentioned in appendix, the paper lacks detailed discussion on ethical approval processes or data sharing (e.g., license type, usage restrictions). These are critical for real-world reproducibility and community adoption.
- The paper does not analyze data distribution differences among participating institutions (e.g., demographic or equipment biases). This should be quantified or discussed more explicitly.
- The exact release plan, access procedures, and metadata structure remain unclear. Without these, the practical usability of the dataset is reduced.

**Questions:**

Please refer to the Weakness.

---

> ### Author Response · Authors · 2025-11-21
> **Response to Reviewer TMat**
>
> We really appreciate the reviewer’s thorough feedback with a positive rating, and would like to address the remaining concerns as follows:
>
> > W1. Although data collection and anonymization are mentioned in appendix, the paper lacks detailed discussion on ethical approval processes or data sharing (e.g., license type, usage restrictions). These are critical for real-world reproducibility and community adoption.
>
> We acknowledge the importance of ethical approval, informed consent procedures, and data-sharing governance in ensuring reproducibility and responsible research. All data used in this study were collected in full compliance with the hospital’s internal regulations and ethical guidelines for handling patient and participant information. The dataset employed in this work was provided by the collaborating hospital with explicit authorization for scientific research purposes.
>
> With respect to data sharing, the ownership and governance of the dataset rest with the hospital. Consequently, requests for access to the dataset for research purposes can be directed to the hospital, and the hospital will evaluate and share the data in accordance with its regulations, approval procedures, and confidentiality safeguards.
> If access is approved, the data will be retrieved from a secure cloud environment managed by the hospital (or its authorized provider) and made available only under controlled conditions, ensuring full compliance with data protection, privacy, and security standards.
>
> > W2. The paper does not analyze data distribution differences among participating institutions (e.g., demographic or equipment biases). This should be quantified or discussed more explicitly.
>
> We appreciate the Reviewer’s suggestion to analyze the data distribution differences among participating institutions. In this response, we have restructured and expanded Appendix A.5 (Analysis of Distributional Differences Across Institutions) to address this directly.
> The major revisions include:
> - **Clarification of Demographic Limitations:** We transparently addressed the unavailability of demographic data for Hospital A due to privacy protocols. To compensate, we provided a detailed statistical characterization of the available demographic subset from Hospital B to better define the known population.
> - **Comparative Analysis of Recording Properties:** We introduced a side-by-side comparison of the recording statistics (duration and yearly distribution) for both institutions to visualize temporal and distributional shifts.
> - **Explicit Discussion of Equipment Bias:** We added a dedicated quantification of the hardware differences (sampling rates and channel counts). We further discussed how this "equipment bias" serves as a feature of the dataset, enhancing its ecological validity by mirroring real-world clinical variations.
>
> We believe these updates provide the necessary quantification and discussion of inter-institutional differences while clearly outlining the dataset's composition.
>
> > W3. The exact release plan, access procedures, and metadata structure remain unclear. Without these, the practical usability of the dataset is reduced.
>
> To clarify the metadata structure, we note that M-EEG is organized according to the BIDS standard. At the top level, the dataset is structured according to the BIDS hierarchy, which includes:
> - dataset_description.json: Contains metadata describing the dataset, its authorship, and BIDS compliance.
> - participants.tsv and participants.json: Contain participant-level demographic and group information.
> - phenotype/: Contains clinical laboratory test results and clinical notes in results.tsv and related metadata in results.json.
> - sub-xxxx/: Contain subject-specific data, including an eeg/ subfolder with EEG recordings, associated metadata, channel information, and a sub-xxxx\_scans.tsv file documenting recording timestamps.
>
> In the revision, we have updated the dataset section (line 253) to explicitly state that M-EEG is organized in a BIDS-compatible EEG format with subject-level and session-level metadata. We have provided a small example subset and schema description in the supplementary material. In addition, the Reproducibility Statement now clearly specifies that, upon acceptance, we will release M-EEG, with controlled access via a request form in accordance with our data-use requirements.

---

> > ### Comment · Reviewer_TMat · 2025-11-24
> >
> > The authors’ response has addressed most of the issues I raised initially. Inspired by Reviewer oVMU, I noticed that the authors have added more discussion of related work in Appendix A.8, which I appreciate very much. In particular, there are references about multimodal and cross-domain EEG research, which reminds me of the Brant series of studies (including Brant, Brant-2, and Brant-X) where Brant-X jointly studies EEG with other physiological signals. Therefore, I suggest that the authors consider including these works in that section to provide a more complete and comprehensive review of the literature.

---

> > > ### Author Response · Authors · 2025-11-28
> > > **Follow-up**
> > >
> > > Thank you for your thoughtful comment and for pointing us to the Brant series. We have extended Appendix A.7 (appendix A.8 in the previous manuscript) to incorporate these works. Specifically, in lines 1224-1233, we summarized Brant, Brant-2, and Brant-X, and add a new comparison table (Table 15) to clearly position our contribution. As summarized in the table below, **Brant** scales foundation models to intracranial SEEG by pretraining exclusively on a large private SEEG cohort, targeting invasive neural recordings rather than scalp EEG. **Brant-2** extends this line of work by training a unified backbone on both SEEG and EEG (private SEEG + TUEG), but still operates **within a single-modality** neural signal space and does not explore explicit multimodal alignment. **Brant-X** moves toward multimodality by jointly modeling EEG with other physiological signals (EOG, ECG, EMG) on CAP, ISRUC, and HMC, focusing on **cross-signal alignment between biosignals rather than fusion multiple modalities**.
> > >
> > > In contrast, our work benchmarks the integration of scalp EEG with blood-based biomarkers for neurological disorder diagnosis, which is not addressed in the Brant series.
> > >
> > > We hope this clarifies our positioning relative to these important works. If you find that your remaining concerns are addressed, we would be very grateful if you could consider an updated score; we are happy to further revise if needed.
> > >
> > > |**Papers**|**Modalities of each sample**|**Datasets**|**Task**|
> > > |-|-|-|-|
> > > |Brant|Only SEEG|a private SEEG dataset|Towards foundation models for intracranial neural signal|
> > > |Brant-2|either SEEG or EEG|a private SEEG dataset, TUEG|Towards foundation models for brain signals|
> > > |Brant-X|either EEG, EOG, ECG, or EMG|CAP, ISRUC, and HMC|Multimodal **alignment**|
> > > |**Ours**|EEG, lab values and clinical notes|M-EEG, T-EEG, TUEG, NMT-Scalp|Multimodal EEG **fusion**|

---

### Official Review · Reviewer_6AaG · 2025-11-01

**Soundness:** 3
**Presentation:** 2
**Contribution:** 2
**Rating:** 4
**Confidence:** 4

**Summary:**

This work tackles the fragmentation and US-centric bias of current EEG and the unimodal focus of existing EEG foundation models. This work introduces M-EEG, which is a large non-US clinical EEG dataset with 6081 patients and paired blood biomarkers and clinical notes and unifies major public EEG datasets into a standardized pretrianing corpus (P-EEG) + a task suite for fair downstream evaluation (T-EEG). Under identical pipelines, the authors benchmark leading EEG foundation models and show that adding regionally diverse data improves out-of-region robustness and that lightweight multimodal fusion of blood tests with EEG yields clear gains on early neurological diagnosis.

**Strengths:**

I found the paper original because the authors built the largest non-US, multimodal clinical EEG set with paired blood tests and notes. It also defines a unified pretrain corpus and a clear downstream suite for fair Foundation model comparison. The quality is also fair, as the data are large, standardized, and used in controlled tests that check both in-region stability and out-of-region transfer. Also, I found the clarity good as the work opens with a concise figure-level overview of the dataset, benchmark and multimodal path, then explains P-EEG and T-EEG plainly. Finally, I think the significance is good as the authors add M-EEG keeps or improves in-region results and clearly lifts out-of-region robustness, and simple EEG+blood fusion shows large gains on Alzheimer's risk prediction.

**Weaknesses:**

There are a couple of concerns that I would like to raise.

(1) From my understanding, the multimodal evidence in this work is narrow, as the dataset includes both blood tests and clinical notes, yet the modelling and results use only blood, and no text modality is evaluated. I was thinking maybe providing a small text-EEG fusion baseline or even an ablation study would better support your multimodal claim.

(2) Also, the linking of labs/notes to the EEG is by subject ID, and there is no stated time window around the EEG, which can mix pre-/post information. What the authors think of spelling out the windowing rule and sharing a sensitivity check with tighter and looser windows.

(3) For downstream benchmarking, heterogeneous datasets are linearly mapped to a 19-channel montage. This harmonization is practical but may distort cross-dataset comparisons. Maybe a sensitivity experiment without mapping (if possible) or with alternative mappings could help, or at least, I'd like to hear the authors' thoughts on it

**Questions:**

I'd encourage the authors to check the weaknesses part first, and here, I have a couple of questions and would appreciate the authors' feedback.

(1) I was wondering do the EEG+blood gains remain after controlling for lab availability and ordering patterns, for example, when you compare only cases with a common lab panel or matched draw times, or could the model be using care-path shortcuts instead of physiology? I think it'd be interesting for the community to know that.

(2) I am also curious to know how you rule out site or device fingerprints as the source of out-of-region gains, given montage templates, amplifier noise, and 50 vs 60 Hz line features can differ by hospital, and would a leave-device-out or site-confusion check share a different story? It's a bit confusing.

(3) My final question is that are the reported improvements are stable across random seeds and alternative splits, and statistically reliable under a patient-level bootstrap with corrections across tasks, so that the conclusion does not hinge on a single partition?

I'm really looking forward to the rebuttals and appreciate the authors' hard work. I would be happy to modify the score if the answers from the authors are convincing.

---

> ### Author Response · Authors · 2025-11-21
> **Response to Reviewer 6AaG (part 1/2)**
>
> We would like to thank the Reviewer for the detailed feedback and address the Reviewer’s question as follows:
>
> > W1. From my understanding, the multimodal evidence in this work is narrow, as the dataset includes both blood tests and clinical notes, yet the modelling and results use only blood, and no text modality is evaluated. I was thinking maybe providing a small text-EEG fusion baseline or even an ablation study would better support your multimodal claim.
>
> We thank the Reviewer for this suggestion. In the revision, we have added Table 12 (line 1070), where we **fuse EEG with text** derived from brain MRI reports for the subset of patients with MRI, providing an explicit EEG-text fusion baseline. The effect of fusing EEG with text is shown in the table below. Without textual information, the models perform only slightly better than random guessing; **once text is incorporated, their performance improves substantially**, with CBraMOD gaining 17.66% and EEGPT gaining 31.97% in B. Accuracy. In line with the related comment from Reviewer 9JfV, we have also **added additional neurological disorder experiments** in Table 5 to further illustrate the benefits of multimodal fusion. The results further show that incorporating blood-based biomarkers yields consistent gains across architectures and neurological disorders. **These results support our multimodal claim.**
>
> | Tasks | Model |  | B. Acc. |  | AUPR |  | AUROC |  |
> |-|-|-|-|-|-|-|-|-|
> | | | | Perf. | Gain (%) | Perf. | Gain (%) | Perf. | Gain (%) |
> | **Epilepsy** | CBraMOD | w/o Text | 0.5282 | | 0.4317 | | 0.5550 | |
> | | | w/ Text | 0.6215 | **17.66** | 0.5846 | **35.42** | 0.6233 | **12.31** |
> | | EEGPT | w/o Text | 0.5120 | | 0.4058 | | 0.5056 | |
> | | | w/ Text | 0.6757 | **31.97** | 0.6783 | **67.15** | 0.7194 | **42.29** |
>
> > W2. Also, the linking of labs/notes to the EEG is by subject ID, and there is no stated time window around the EEG, which can mix pre-/post information. What the authors think of spelling out the windowing rule and sharing a sensitivity check with tighter and looser windows.
>
> Currently, **our dataset contains only single-day EEG recordings per patient**, without multi-day follow-up sessions. **Laboratory results and clinical notes are collected on the same calendar day as the EEG**. Each labs/notes modality is linked to the corresponding EEG via the subject ID, and the acquisition date is obtained from the acq_time field in the sub-xxxx_scans.tsv files, which follows the YYYY-MM-DDTHH:MM:SS format. Since timestamps are effectively only available at the day level in the current version, we are not able to conduct experiments with tighter/looser temporal windows. We have updated the paper accordingly (Line 260 in the revised version).
>
> > W3. For downstream benchmarking, heterogeneous datasets are linearly mapped to a 19-channel montage. This harmonization is practical but may distort cross-dataset comparisons. Maybe a sensitivity experiment without mapping (if possible) or with alternative mappings could help, or at least, I'd like to hear the authors' thoughts on it
>
> We thank the Reviewer for the comment. In our paper, the phrase “linear channel mappings are applied when necessary to align with the pretrained 19-channel montage” is to convey that whether channel mapping is applied depends on the specific requirements of each foundation model. For example, in **CBraMOD**, the encoder is designed to operate directly on the native channel configuration of each dataset, and therefore **no linear mapping is needed.**
> In contrast, EEGPT is pretrained on a fixed 19-channel subset, thus datasets whose channels do not match this space, we apply a linear mapping to 19-channel montage before feeding the signals into the encoder.
>
> To address the Reviewer’s concern, we conducted additional experiments with EEGPT in which all datasets were fed in their native channel configuration, without any mapping to 19 channels. We used two datasets: Sleep-EDFx (2 channels) and TUAB (23 channels). For Sleep-EDFx, signals were passed directly to the encoder and use existing channels embeddings; for TUAB, we added 4 extra channel embeddings.
>
> The results in the table below, indicate that the impact of linear mapping is minimal. For Sleep-EDFx, the performance with linear mapping is slightly better than without it; for TUAB, the performance drop is marginal (approximately 1%). These findings confirm that the mapping strategy does not materially affect the conclusions of our paper, and we have incorporated these results into the revised version (Tab. 9).
>
> | Tasks | Linear map | B. Acc.  | | Cohen's Kappa / AUPR | | Weighted F1 / AUROC | |
> |-|-|-|-|-|-|-|-|
> | | | Perf. | Diff. (%) | Perf. | Diff. (%) | Perf. | Diff (%) |
> | **TUAB** | Yes | 0.8018 | | 0.8808 || 0.8826 |
> | | No | 0.8136 | **+1.47** | 0.8946 |**+1.57**| 0.8916 | **+1.02**|
> | **Sleep-EDFx** | Yes | 0.6585 | | 0.5963 | | 0.6976 |
> | | No | 0.6009 | **-8.75** | 0.5556 | **-6.83** |0.6574 |**-5.76**|

---

> ### Author Response · Authors · 2025-11-21
> **Response to Reviewer 6AaG (part 2/2)**
>
> > Q1. I was wondering do the EEG+blood gains remain after controlling for lab availability and ordering patterns, for example, when you compare only cases with a common lab panel or matched draw times, or could the model be using care-path shortcuts instead of physiology? I think it'd be interesting for the community to know that.
>
> We thank the Reviewer for raising this important point. If we understand correctly, the concern is that: *an Alzheimer’s patient might be routed through a characteristic care path (e.g., being ordered a specific set of lab tests), and the model could then learn to use the fact that a particular test was ordered as a proxy for disease status rather than relying on physiology.*
>
> If our interpretation is correct, then the answer is as follows:
>
> In our current experiments, we explicitly control for lab availability and ordering patterns in the following sense: the blood-based features fed into the model are restricted to lab values that are available for all subjects. Thus, every patient, both those with high Alzheimer’s risk and those without, undergoes the same panel of tests, and the model only sees the resulting values (not which tests were ordered or when). This design removes the “test ordered vs. not ordered” shortcut and forces the model to rely on physiological differences reflected in the shared lab panel. We have updated Appendix A.3 to clearly list (and highlight in blue) the exact lab values used in our experiments. We agree that the broader problem of missing modalities and partially observed lab panels is important, but a full treatment of missing-modality modeling is beyond the scope of this work; here, our goal is to demonstrate that minimally invasive modalities such as standard blood-based biomarkers do provide meaningful signal for early diagnosis and risk stratification.
>
> >Q2. I am also curious to know how you rule out site or device fingerprints as the source of out-of-region gains, given montage templates, amplifier noise, and 50 vs 60 Hz line features can differ by hospital, and would a leave-device-out or site-confusion check share a different story? It's a bit confusing.
>
> We thank the Reviewer for raising this point and would like to clarify our understanding. If we understand correctly, the concern is that: *differences in site or device (e.g., montage, amplifier noise, or acquisition settings) might allow the model to exploit “fingerprints” of a particular hospital or device, essentially learning that certain labels tend to be recorded on a specific device or at a specific site, rather than learning physiologically meaningful patterns.*
>
> If our interpretation is correct, then the answer is as follows:
>
> In our setting, however, all downstream tasks reported in Table 4 are acquired on the same device within the same hospital, and the corresponding device configurations are detailed in Table 6. This substantially limits the possibility that the observed effects are driven by site-specific or device-specific shortcuts rather than by the underlying EEG signal.
>
> >Q3. My final question is that are the reported improvements are stable across random seeds and alternative splits, and statistically reliable under a patient-level bootstrap with corrections across tasks, so that the conclusion does not hinge on a single partition?
>
> Yes, the reported improvements do not rely on a single partition.
> For each downstream dataset, we use patient-level splits: either k-fold cross-validation or leave-one-subject-out (LOSO), depending on dataset size. In both cases, we cycle the folds in a round-robin manner, so that each fold (or subject) serves as the test set exactly once and performance is aggregated over all resulting test folds. We updated Appendix A.1 (highlighted in blue) to describe the preprocessing and cross-validation strategy for each downstream task in detail (Lines 780).
>
> We hope our responses have fully addressed your remaining concerns, and we would sincerely appreciate your consideration in raising your score if no further issues persist. If you have any further questions or points that require additional clarification, please feel free to let us know. We would be happy to continue the discussion and provide any additional details.

---

> > ### Author Response · Authors · 2025-11-28
> > **Follow-up**
> >
> > Dear Reviewer 6AaG,
> > We sincerely hope that our rebuttal has clarified the points you raised and satisfactorily addressed your concerns.
> >
> > Should you have any further questions or feel that any issue requires additional explanation, please let us know.
> >
> > We would be more than happy to provide further clarification or engage in additional discussion.

---

### Official Review · Reviewer_9JfV · 2025-11-01

**Soundness:** 3
**Presentation:** 2
**Contribution:** 4
**Rating:** 8
**Confidence:** 3

**Summary:**

This paper addresses key limitations in EEG foundation modeling: the lack of standardized benchmarks, the strong US-centric bias of existing datasets, and the unimodal (EEG-only) nature of current models.

The authors make three main contributions:

1. M-EEG Dataset: They introduce M-EEG, a new, large-scale (6,081 patients, 1,170 hours) multi-institutional clinical EEG dataset collected from two hospitals outside the US. This is the largest non-US dataset by subject count and uniquely features a multimodal subset with paired EEG, blood-based biomarkers (BBB), and clinical notes.
2. Standardized Benchmarking (P-EEG & T-EEG): They create a new benchmarking framework. This includes P-EEG, a unified pretraining corpus created by harmonizing their new M-EEG dataset with existing public datasets (TUEG and NMT Scalp), and T-EEG, a diverse benchmark of seven downstream task-oriented datasets. This framework enables the first rigorous, fair comparison of state-of-the-art EEG foundation models (CBraMOD and EEGPT).
3. Multimodal Validation: They demonstrate the value of multimodality by designing a multimodal model that fuses EEG and blood biomarkers using cross-attention. Experiments on the PEARL dataset (using models pretrained on P-EEG) show that adding blood biomarkers significantly improves diagnostic accuracy for Alzheimer's risk prediction, achieving up to a 27.64% relative gain in balanced accuracy over an EEG-only baseline.

**Strengths:**

1. Significant Data Contribution: The primary strength is the introduction of the large-scale EEG dataset. This directly addresses the critical lack of geographic and demographic diversity in existing EEG corpora (e.g., TUH, HEEDB).

2. Novel Multimodality: M-EEG is uniquely multimodal, including paired EEG, blood-based biomarkers (BBB), and clinical notes for a subset of patients. This is a significant contribution that opens the door for a new class of multimodal foundation models for neurology, moving beyond EEG-only signals to incorporate other clinically vital, minimally-invasive data.

3. First Standardized EEG FM Benchmark: The paper introduces a comprehensive and standardized benchmarking framework (P-EEG for pretraining, T-EEG for downstream tasks). This is a high-quality contribution that allows for a rigorous and fair comparison of SOTA EEG FMs under consistent protocols. This addresses a major limitation of prior work where fair comparison was impossible.

4. Strong Empirical Evidence and Key Insights: The paper provides strong, well-supported evidence for two important claims:
   - Regional Diversity Matters: The experiments clearly show that pretraining with the non-US M-EEG data substantially improves model generalization to out-of-region (OOD) data while maintaining performance on in-region tasks (Tables 3 & 4).
   - Multimodality is High-Value: The paper demonstrates that integrating blood biomarkers with EEG leads to significant performance gains in a clinical task. This is a significant result for the clinical utility of FMs.

**Weaknesses:**

1. Dataset Naming: The paper is severely undermined by a major, recurring inconsistency in the name of its own dataset. The title, abstract, and Section 3.1 call it "M-EEG". However, it is repeatedly referred to as "VEEG" throughout the paper, including in Section 2, Figure 1 (and its caption), Section 3.2, and Section 4.3. This is an extremely confusing and careless error that must be fixed.

2. Misleading Wording in Abstract: The abstract contains two statements that are either incorrect or misleading based on the paper's own contents:

   - It claims P-EEG "unify(s) all existing public EEG datasets". This is factually incorrect. Section 3.2.1 clearly states P-EEG consists of only three datasets (TUEG, NMT Scalp, M-EEG) and explicitly excludes major corpora like HEEDB and SHHS. This should be rephrased to "unifies several key public datasets."
   - It states, "using our multimodal EEG dataset... achieving a 27.64% gain". This implies the multimodal evaluation was performed on M-EEG. However, the experiment (Section 4.4, Table 5, Table 11) was conducted on the external PEARL dataset. While the model was pretrained on P-EEG (which contains M-EEG), the evaluation dataset for this specific claim was PEARL. This wording should be clarified.

3. Confusing/Redundant Results Presentation: The presentation of the multimodal results in the appendix is cluttered. There are four separate tables (Tables 9, 10, 11, and 12) that show different slices of the same core experiment (original vs. P-EEG-pretrained checkpoints, concat vs. attention fusion, vs. BBB-only). This can be consolidated into one or two clear, summary tables that directly compare the most important conditions.

**Questions:**

1. Please clarify the correct name of the dataset you are introducing. The paper uses "M-EEG" and "VEEG" interchangeably (e.g., Title/Abstract vs. Figure 1/Section 2). This is a major point of confusion and must be corrected.

2. The abstract claims P-EEG unifies "all" existing datasets, but Section 3.2.1 states it's composed of TUEG, NMT-Scalp, and M-EEG, and excludes HEEDB. Can you confirm this and that you will correct the abstract to reflect the actual composition?

3. The abstract's claim of a "27.64% gain" appears to be based on experiments on the external PEARL dataset (Table 5/11), not on a downstream task within the new M-EEG dataset. Could you please clarify this? Is there a multimodal downstream task (e.g., for AD risk) evaluated on M-EEG itself, or is M-EEG's multimodal component primarily offered as a resource for future work?

4. The multimodal fusion (Section 4.4, Appendix A.3) uses blood-based biomarkers (BBB). Can you provide more detail on which specific biomarkers from the blood tests were used? Is it the full set of available lab values, or a curated subset (e.g., complete blood count, inflammatory markers) known to be relevant to neurodegeneration?

**Details Of Ethics Concerns:**

The authors state that the data was "fully de-identified before release" and "anonymized", following standard procedures for handling sensitive patient data. So no ethics review is needed.

---

> ### Author Response · Authors · 2025-11-20
> **Response to Reviewer 9jfV**
>
> We truly appreciate the Reviewer's thoughtful feedback and acceptance rating. We would like to address the Reviewer's questions below:
>
> > W1 & Q1. Please clarify the correct name of the dataset you are introducing. The paper uses "M-EEG" and "VEEG" interchangeably. This is a major point of confusion and must be corrected.
>
> The correct dataset name is "M-EEG", as stated in the title and abstract. All occurrences of "VEEG" in the main text were typographical errors referring to the same dataset. We have corrected all instances to "M-EEG" (Lines 67, 71, 73, 75, 269, 341, 1242, Fig. 1 in the revised manuscript). This change does not affect any analyses.
>
> > W2.1 & Q2. The abstract claims P-EEG unifies "all" existing datasets, but Section 3.2.1 states it's composed of TUEG, NMT-Scalp, and M-EEG, and excludes HEEDB. Can you confirm this and that you will correct the abstract?
>
> The Reviewer is correct that P-EEG is composed of TUEG, NMT-Scalp, and M-EEG, and excludes HEEDB. The phrase “all existing datasets” in the original abstract was overly strong and may have been misleading. We have updated the abstract to accurately describe P-EEG, stating that we unify “several key public EEG datasets” (Line 24 in the revised manuscript).
>
> > W3. Confusing/Redundant Results Presentation: The presentation of the multimodal results in the appendix is cluttered. There are four separate tables (Tab. 9, 10, 11, and 12) that show different slices of the same core experiments. This can be consolidated into one or two clear, summary tables.
>
> We thank the Reviewer for this suggestion. In the revision, we have consolidated the original four tables (9-12) into two clearer summary tables: Tab. 10 (original checkpoints) and Tab. 11 (P-EEG checkpoints). These tables unify the multimodal results, highlighting (i) consistent gains from adding blood-based biomarkers and (ii) ablations supporting our attention-based fusion.
>
> > W2.2 & Q3. The abstract's claim of a "27.64% gain" appears to be based on experiments on the external PEARL dataset (Tab. 5), not on a downstream task within the new M-EEG dataset. Could you please clarify this? Is there a multimodal downstream task evaluated on M-EEG itself, or is M-EEG's multimodal component primarily offered as a resource for future work?
>
> The reported 27.64% gain in the abstract indeed comes from the Alzheimer’s risk prediction experiment on the external PEARL dataset, where we aim to show the potential of combining EEG-FMs with minimally invasive biomarkers for early risk screening. In the revision, we have clarified explicitly by revising the abstract (line 27) and splitting the original Sec. 4.4 (Impacts of Multimodality Data) into two subsections: 4.4.1 Experimental Results on PEARL and 4.4.2 Experimental Results on M-EEG for Neurological Disorders Prediction, where we additionally curate and evaluate 3 downstream multimodal tasks directly from M-EEG (epilepsy, transient ischemic attack (TIA), and Parkinson). We now include these new M-EEG evaluations in the table below and update the manuscript accordingly in Tab.5, App. A1 and App. A4, thereby strengthening our evaluation and explicitly benchmarking M-EEG’s multimodal components rather than offering them only as a resource for future work.
>
>
> | Tasks | Model |  | B. Acc. |  | AUPR |  | AUROC |  |
> |-|-|-|-|-|-|-|-|-|
> | | | | Perf. | Gain (%) | Perf. | Gain (%) | Perf. | Gain (%) |
> | **Epilepsy** | CBraMOD | w/o BBB | 0.5248 | | 0.4262 | | 0.5142 | |
> | | | w/ BBB | 0.6280 | **19.67** | 0.5457 | **28.04** | 0.7011 | **36.35** |
> | | EEGPT | w/o BBB | 0.5144 | | 0.4126 | | 0.5494 | |
> | | | w/ BBB | 0.6306 | **22.59** | 0.5801 | **40.60** | 0.6942 | **26.36** |
> | **TIA** | CBraMOD | w/o BBB | 0.5266 | | 0.4003 | | 0.6234 | |
> | | | w/ BBB | 0.5680 | **7.86** | 0.6385 | **59.51** | 0.7402 | **18.74** |
> | | EEGPT | w/o BBB | 0.5446 | | 0.5269 | | 0.5776 | |
> | | | w/ BBB | 0.6263 | **15.00** | 0.5654 | **7.31** | 0.6219 | **7.67** |
> | **Parkinson** | CBraMOD | w/o BBB | 0.5556 | | 0.7850 | | 0.7396 | |
> | | | w/ BBB | 0.6667 | **20.00** | 0.9681 | **23.32** | 0.9519 | **28.70** |
> | | EEGPT | w/o BBB | 0.6157 | | 0.7755 | | 0.8153 | |
> | | | w/ BBB | 0.7667 | **24.53** | 0.9464 | **22.04** | 0.9505 | **16.58** |
>
> >Q4. The multimodal fusion (Sec. 4.4, App. A.3) uses blood-based biomarkers (BBB). Can you provide more detail on which specific biomarkers from the blood tests were used? Is it the full set of available lab values, or a curated subset (e.g., complete blood count, inflammatory markers) known to be relevant to neurodegeneration?
>
> We have clarified the exact BBB used for multimodal fusion (Lines 1003 and 1050 in the revised manuscript). For PEARL, the BBB comprises all routinely available lab values, including complete blood count and a standard lipid panel, rather than a hand-picked neurodegeneration-specific subset. For the M-EEG cohort, we use the same markers plus electrolytes, liver and renal markers, uric acid, calcium, glucose, and HbA1c.

---

> > ### Comment · Reviewer_9JfV · 2025-11-24
> >
> > I appreciate the detailed clarifications from the authors, which have addressed my specific concerns. Nonetheless, I will maintain my current rating as I believe it remains the most appropriate given the current state of the manuscript.
> >
> > I agree with Reviewer oVMU that the presentation of the paper could be significantly improved. The current focus is diluted, which leads to the importance and details of the M-EEG dataset being understated. To better align the paper with the Datasets and Benchmarks track, I strongly encourage the authors to restructure the paper to center on the contribution of the new dataset. Specifically, I recommend the following revisions:
> > 1. Relocate the critical description and metadata of M-EEG to main text: Essential metadata, including demographic and site-specific information, should be prominent rather than hidden in the appendix.
> > 2. De-emphasize the multi-modal model architecture: The architecture should serve primarily to validate the utility of the dataset. Please avoid framing it as a novel architectural contribution.
> > 3. Strengthen the argument on regional diversity: Since a key strength of this work is presenting the largest non-US dataset, this should be highlighted. For example, demonstrate more geographic diversity in Section 3.1 and, if possible, compare the demographics against US-based datasets to underscore the gap this dataset fills.
> > 4. Highlight the dataset usability: Detail the dataset structure and usability in the main text; specifically, compliance with BIDS standards should be explicitly highlighted.
> > 5. Clarify the primary contribution: I recommend positioning M-EEG as the primary contribution of novel resource. Frame P-EEG and T-EEG as standardization efforts to facilitate the benchmarking of M-EEG, ensuring they do not distract from the novelty of the newly collected data.

---

> > > ### Author Response · Authors · 2025-11-28
> > > **Follow-up**
> > >
> > > We would like to thank the Reviewer for the constructive comments and for the suggestions to improve the presentation of our paper. In the revised manuscript, we have made the following changes:
> > >
> > > > 1. Relocate the critical description and metadata of M-EEG to main text: Essential metadata, including demographic and site-specific information, should be prominent rather than hidden in the appendix.
> > >
> > > We have moved the essential metadata of M-EEG to the main text. In particular, we introduced a new Table 2 and Figure 2 (line 270) that present the age and gender distributions directly in the main text instead of the appendix. We also summarized site-specific information in a new Table 3 (line 283).
> > > > 2. De-emphasize the multi-modal model architecture: The architecture should serve primarily to validate the utility of the dataset. Please avoid framing it as a novel architectural contribution
> > >
> > > We have revised both the abstract and the introduction to clarify this point. In the abstract (line 27), we now clarify that we “configure and evaluate multimodal diagnostic models based on existing EEG foundation architectures.” In the introduction (lines 93 - 98), we further emphasize that we “adapt existing EEG foundation architectures to a multimodal setting for neurological diagnosis,” rather than proposing a new architecture.
> > >
> > > > 3. Strengthen the argument on regional diversity: Since a key strength of this work is presenting the largest non-US dataset, this should be highlighted. For example, demonstrate more geographic diversity in Section 3.1 and, if possible, compare the demographics against US-based datasets to underscore the gap this dataset fills.
> > >
> > > At line 253, we explicitly clarified that all patients in our dataset are recruited from a country geographically distant from the US. We also point to Figure 2 and Table 2, which show that our newly curated multimodal dataset spans a wide range of ages and includes substantial representation of both male and female patients, contributing a large volume of EEG recordings paired with other modalities. Together, these additions highlight how M-EEG expands and diversifies the current public corpus toward an under-represented region.
> > >
> > > > 4. Highlight the dataset usability: Detail the dataset structure and usability in the main text; specifically, compliance with BIDS standards should be explicitly highlighted.
> > >
> > > At line 301, we have moved and expanded the description of the dataset structure and standardization, including an explicit statement that the dataset follows BIDS standards, into the main text to emphasize usability.
> > >
> > > > 5. Clarify the primary contribution: I recommend positioning M-EEG as the primary contribution of novel resources. Frame P-EEG and T-EEG as standardization efforts to facilitate the benchmarking of M-EEG, ensuring they do not distract from the novelty of the newly collected data.
> > >
> > > Following this suggestion (and in line with Reviewer oVMU’s comments), we have reorganized the list of main contributions into two key points (lines 081-099). The first now highlights M-EEG as the primary novel resource together with our associated benchmarking effort, while the second focuses on the multimodal evaluation for neurological diagnosis. Consistently, P-EEG and T-EEG are now explicitly framed as standardized corpora that support benchmarking and amplify the utility of M-EEG (lines 209 and 215).
> > > Once again, we thank the Reviewer for the acceptance rating and for the thoughtful feedback that helped us improve the clarity and presentation of the manuscript. We would be happy to further discuss any remaining concerns.

---

### Author Response · Authors · 2025-12-04
**Summary of the revised manuscript**

Dear AC and Reviewers, We extend our sincere gratitude to the Reviewers for providing valuable feedback. Your insights have been instrumental in improving our work.

We have carefully addressed all comments and suggestions from the Reviewers, ensuring that their feedback is thoroughly incorporated into both the revised manuscript and the Appendix.

All changes in the revised manuscript and supplementary materials are highlighted in blue. We believe the updated submission reflects significant improvements, effectively clarifying and resolving the concerns previously raised.

Below, we summarize the key changes made to our revised manuscript:
1. **Dataset’s name clarification** (Reviewer **9jfV**)
- We have standardized the dataset name to “**M-EEG**” and updated all previous occurrences of “VEEG” accordingly (**Lines 68, 72, 74, 76, 318, 378, and Fig. 1**).
2. **Abstract and Introduction Revision** (Reviewer **9jfV, oVMU**)
- The Introduction has been revised to highlight the distinction between our approach and existing methods. In particular, we now clearly emphasize that M-EEG is the **first non-US, multi-institutional, multimodal EEG dataset**, constitutes **the largest non-US EEG cohort by subject count, enriches the diversity of EEG pretraining resources**, and supports a **multimodal benchmark for neurological diagnosis**.
3. **Additional Experiments**

We performed several new experiments to better highlight our contributions. These experiments include:
- Analyzing the impact of combining blood-based biomarkers with EEG for neurological diagnosis on M-EEG, covering three major neurological disorders (epilepsy, transient ischemic attack, and Parkinson’s disease), in addition to the original Alzheimer’s risk prediction experiment on the PEARL dataset (**Table 7**) (Reviewer **9jfV, 6AaG**),
- Analyzing the impact of incorporating free-text clinical notes alongside EEG for epilepsy prediction on M-EEG (**Lines 1060-1079**) (Reviewer **6AaG**),
- Adding an ablation study with linear mapping on EEGPT (**Lines 942-949**) (Reviewer **6AaG**),
4. **Enhanced Explanations and Presentation**
- To improve clarity and to highlight our contributions, we have relocated the critical description, including the dataset demographics, usability and metadata of M-EEG to the main text. This also strengthen the argument on regional diversity (**Lines 252-255; Lines 261-314; Tables 2-3; Figure 2**) (Reviewer **9jfV, TMaT**),
- We added details about the specific biomarkers used and the collected time stamps of their collection (**Line 269; Lines 455-457; Lines 1003-1010; Lines 1049-1059**) (Reviewer **9jfV, 6AaG**),
- To improve reproducibility, we described the cross-validation protocol for each downstream task and clarified that our evaluation does not rely on a single train-test partition (**Appendix A.1**) (Reviewer **6AaG**).
5. **Ethics Concerns Resolve**
- We addressed the ethics concerns raised by adding information on dataset ownership, governance, and the access protocol to M-EEG (**Reproducibility Statement**) (Reviewer **TMaT**)
6. **Inclusion of Relevant Works**
- We have incorporated all relevant studies highlighted by Reviewer **oVMU, TMaT** into the revised manuscript and provided a clear explanation of how our contribution differs from existing works (**Table 15; Appendix A.7**)

We hope that our detailed rebuttal and revised manuscript satisfactorily address the concerns raised during the review process, and that this summary will help the Area Chair and all Reviewers clearly understand what has already been discussed and improved.

---

### Meta-Review · Area_Chair_7YVJ · 2026-01-04

**Summary:**

This submission introduces a large-scale, non-US clinical scalp EEG dataset with paired multimodal information, including blood-based biomarkers and clinical notes (mainly derived from brain MRI reports), and presents the first benchmarking effort targeting EEG foundation models with multimodal capabilities. Reviewers generally agree that the dataset itself is the strongest contribution of the paper, addressing a clear gap in geographic diversity and enabling standardized evaluation of EEG foundation models in clinically relevant settings.

However, the technical contributions beyond the dataset are relatively limited. As noted by multiple reviewers (including Reviewers 6AaG and oVMU), the proposed multimodal fusion that relies on cross-modality attention mechanisms are well established in prior multimodal learning literature, and the model architecture is not positioned as a novel technical contribution (as additionally commented by Reviewer 9JfV). The empirical gains from multimodal fusion, while consistent, are modest in some settings, and in certain cases only marginally above a random baseline (as illustrated in the author's response to Reviewer 6AaG (part 1/2)), suggesting limited exploitable structure in the EEG signals for the evaluated tasks.

More broadly, reviewers found that the experimental results provide limited new insight for the community. While demonstrating that incorporating auxiliary clinical modalities improves predictive performance is useful for validation, this finding is already well understood and does not substantially advance EEG-specific modeling knowledge or clinical practice. In addition, the benchmarking of EEG foundation models remains narrow, as comparisons are restricted to only two existing methods (EEGPT and CBraMOD), limiting the strength of conclusions regarding model design or generalization.

In summary, this work offers a valuable dataset and a reasonable first step toward multimodal benchmarking for EEG foundation models, but the lack of technical novelty and the limited depth of empirical insights reduce its overall impact. The contribution may be more appropriate for a datasets&benchmarks–focused venue or track, unless the authors can further sharpen the scope, expand the benchmarking coverage on baselines, and strengthen the analytical insights derived from the benchmark.

Based on the reviews and rebuttal discussion, I do not recommend acceptance of this paper.

**Reviewer Concerns:**

Concerns addressed by the rebuttal:
* Dataset naming inconsistency (VEEG)
* Overstated claims in the abstract
* Multimodal evidence being too narrow (Reviewer 6AaG)
* Concerns about shortcut learning (lab availability, site or device effects) and statistical robustness
* Ethics, data governance, metadata structure, and usability
* Related work coverage

Concerns that remain partially outstanding:
* Overall presentation and focus: Despite improvements, multiple reviewers (notably 9JfV and oVMU) still view the paper as trying to do too much, with the core dataset contribution potentially diluted by extensive modeling and benchmarking content.
* Strength of novelty compared to prior benchmarks.
* Perceived maturity of contributions. The core contribution of the paper lies in the dataset, while the technical contributions are limited, as also pointed out by Reviewers 9JfV and oVMU.

**Reviewer Scores:**

Based on the discussion and the rebuttal, I do not expect any reviewer to have materially changed their score had they been able to participate fully in the discussion.

Reviewer 9JfV: No change (8). The reviewer has responded to the author's rebuttal and given some further suggestions to strenghten the paper.

Reviewer oVMU: No change (4). The rebuttal did not materially alter the reviewer's assessment that the contribution is primarily dataset-driven with insufficient methodological novelty.

Reviewer 6AaG: May slightly lead to changing to 6. Although additional experiments improved clarity, the scope and depth of insights were not significantly expanded.

Reviewer TMat: No change (6). Clarifications on dataset governance and related work were helpful but did not affect the overall evaluation.

---

### Decision · Program_Chairs · 2026-01-26

Reject